# Molecular Mechanism of Autophagosome–Lysosome Fusion in Mammalian Cells

**DOI:** 10.3390/cells13060500

**Published:** 2024-03-13

**Authors:** Po-Yuan Ke

**Affiliations:** 1Department of Biochemistry & Molecular Biology, Graduate Institute of Biomedical Sciences, College of Medicine, Chang Gung University, Taoyuan 33302, Taiwan; pyke0324@mail.cgu.edu.tw; Tel.: +886-3-211-8800 (ext. 5115); Fax: +886-3-211-8700; 2Liver Research Center, Chang Gung Memorial Hospital, Taoyuan 33305, Taiwan

**Keywords:** autophagy, autophagosome, lysosome, autolysosome, autophagosome–lysosome fusion

## Abstract

In eukaryotes, targeting intracellular components for lysosomal degradation by autophagy represents a catabolic process that evolutionarily regulates cellular homeostasis. The successful completion of autophagy initiates the engulfment of cytoplasmic materials within double-membrane autophagosomes and subsequent delivery to autolysosomes for degradation by acidic proteases. The formation of autolysosomes relies on the precise fusion of autophagosomes with lysosomes. In recent decades, numerous studies have provided insights into the molecular regulation of autophagosome–lysosome fusion. In this review, an overview of the molecules that function in the fusion of autophagosomes with lysosomes is provided. Moreover, the molecular mechanism underlying how these functional molecules regulate autophagosome–lysosome fusion is summarized.

## 1. Introduction

The elimination of unwanted intracellular materials through autophagy critically regulates nutrient recycling and energy regeneration, thus promoting cellular homeostasis [1,2,3]. The dysregulation of autophagy has been shown to potentially be linked to the development of human diseases [3,4,5,6]. Three types of autophagy have been identified: macroautophagy, microautophagy, and chaperone-mediated autophagy (CMA) (Figure 1) [3,7,8,9]. Macroautophagy, the main form of autophagy studied (often referred to as “autophagy”), involves the initial rearrangement of the intracellular membrane for the engulfment of cytosolic materials within double-membrane autophagosomes that ultimately fuse with lysosomes to form autolysosomes, in which acid hydrolases eliminate cargo (Figure 1) [3,9,10,11]. Microautophagy is a degradative process that involves the invagination of the lysosomal membrane to sequester cytoplasmic components within the lysosomal lumen directly and degrade the cargo (Figure 1) [12,13,14,15]. CMA is a lysosomal degradation agent that engulfs proteins containing a pentapeptide motif, the so-called “Lys-Phe-Glu-Arg-Gln” (KFERQ) motif, through the recognition of heat shock protein 70 kDa (HSC70) and docking with lysosomal membrane protein 2A (LAMP2A) on the lysosomal membrane (Figure 1) [16,17,18]. Then, the engulfed cargo is delivered into the lumen of the lysosome and eliminated by lysosomal proteases [16,17,18]. During the process of macroautophagy, the precise control of the fusion between autophagosomes and lysosomes promotes autolysosome maturation (Figure 1) [3,7,8,9]. Several functional molecules that regulate autophagosome–lysosome fusion have been identified in recent years [19,20,21,22,23,24]. Additionally, the improper control of autophagosome fusion with lysosomes has recently been shown to be involved in the pathogenesis of human disorders [19,20,21,22,23,24]. This review summarizes the current knowledge on the molecular mechanism underlying autophagosome–lysosome fusion. Moreover, the functional molecules involved in this fusion process are reviewed.

## 2. Overview of the Autophagy Process

The process of autophagy involves stepwise vacuole biogenesis, including the initial nucleation of the isolation membrane (IM)/phagophore, the elongation and enclosure of the IM/phagophore into a double membranous autophagosome, and the fusion of the autophagosome with the lysosome to form an autolysosome (Figure 2). Multiple core autophagy-related gene (ATG) complexes and kinase signaling pathways pivotally function in each stage of the entire process of autophagy [3,10,11,25,26]. In mammalian cells, an insufficient nutrient supply suppresses the biological activity of the mammalian target of rapamycin (mTOR), a cellular growth-regulated serine/threonine protein kinase, leading to the translocation of the unc-51 like-kinase (ULK) complex, which contains ULK1/2, ATG13, ATG101, and FAK family-interacting protein of 200 kDa (FIP200, also named RB1CC1), from the cytosol to the endoplasmic reticulum (ER) [27,28,29,30]. Then, the ULK complex recruits and phosphorylates the class III phosphatidylinositol-3-OH kinase (PI3KC3) complex, which consists of PI3KC3 (also known as Vps34, Beclin 1, and ATG14), triggering the generation of phosphatidylinositol-3-phosphate (PtdIns(3)P) [26,31,32,33]. The generated PtdIns(3)P subsequently recruits double-FYVE-containing protein 1 (DFCP1) and WD-repeat domain PtdIns(3)P-interacting (WIPI, the mammalian ortholog of ATG18) family proteins to support the emergence of ER-reconstituted IM/phagophores (Figure 2) [26,31,32,34,35]. Additionally, ATG9 and vacuole membrane protein 1 (VMP1) promote the assembly of the IM/phagophore and the dissociation of the IM/phagophore from the ER, respectively [36,37,38,39,40,41,42,43]. In addition to the ER, mitochondria, the plasma membrane, the Golgi apparatus, endosomes, and the mitochondria-associated ER membrane (MAM) also support the membrane source for the nucleation of the IM/phagophores [44,45,46,47,48,49,50,51].

Subsequently, the IM/phagophore elongates into a double-membrane autophagosome through the enzymatic cascades of two ubiquitin-like (UBL) conjugation systems (Figure 2) [30,52,53,54,55,56]. First, ATG7 and ATG10 act as ubiquitination-activating enzyme 1 (E1) and ubiquitin-conjugating enzyme 2 (E2), activating the formation of an ATG12-ATG5 conjugate, which further binds to ATG16, forming an ATG12-ATG5-ATG16L heterotrimeric complex [52,53,57,58,59]. Second, another UBL conjugation cascade of ATG7 (E1), ATG3 (E2), and ATG12-ATG5-ATG16L, a ubiquitin E3 ligase, activates phosphatidylethanolamine (PE) conjugation to the ATG4-cleaved form of ATG8/LC3 family proteins (referred to as ATG8/LC3-I), forming lipidated ATG8/LC3 (ATG8/LC3-PE, referred to as ATG8/LC3-II) [60,61,62,63,64]. PE-conjugated ATG8/LC3 facilitates IM/phagophore expansion to promote the closure of autophagosomes through membrane fusion [55,65,66]. Additionally, the interaction between UV radiation resistance-associated protein (UVRAG) and the PI3KC3 complex promotes the trafficking of endosomes for the maturation of autophagosomes [32,67,68]. The expansion of the IM/phagophore requires the transfer of phospholipids from the ER (Figure 2) [69,70,71,72,73,74,75,76,77,78]. Through binding to ATG8/LC3s and WIPIs [73,79,80], ATG2 can be recruited to the IM/phagophore and act as a lipid channel for the intermembrane transport of phospholipids from the ER to the growing IM/phagophore [72,75,78]. ATG2 can interact with lipid scramblases, including ATG9, VMP1, and transmembrane protein 41B (TMEM41B), to form a heteromeric complex for balancing lipid homeostasis and supporting membrane dynamics within the IM/phagophore (Figure 2) [70,74,77]. In addition to ATG2, vacuolar protein sorting (VPS) 13 may also serve as another lipid transporter for driving IM/phagophore expansion [81,82,83]. After the IM/phagophore expands completely, the endosomal sorting complex required for transport (ESCRT) machinery (containing three subcomplexes: ESCRT-I, ESCRT-II, and ESCRT-III) mediates the closure of the IM/phagophore into mature autophagosomes (Figure 2) [84,85,86,87,88]. The ESCRT-III component, charged multivesicular body protein 2a (CHAMP2A), and the AAA-ATPase VPS4 translocate to the IM/phagophore and promote the sealing of the IM/phagophore for autophagosome completion (Figure 2) [85,88]. In addition, the ESCRT-I subunit, VPS37A, can also be recruited to the IM/phagophore, where it promotes the assembly of CHAMP2 and VPS4, which are required for IM/phagophore closure [86].

Mature autophagosomes may fuse with vacuoles involved within the endolysosomal pathway, including early endosomes, late endosomes/multivesicular bodies (MVBs), and lysosomes. The fusion of autophagosomes to early and late endosomes leads to the formation of nondegradative amphisomes [89,90,91,92,93], which ultimately fuse with lysosomes to form autolysosomes (Figure 1) [20,21,23,24,56,94]. Although the biogenesis of amphisomes through the fusion of autophagosomes with endosomes for autophagic degradation is considered an obligate process, amphisome formation can promote the maturation of autophagosomes and facilitate further fusion of autophagosomes to lysosomes, presumably through the supply of Ras-related (Rab) family proteins and soluble N-ethylmaleimide-sensitive-factor attachment protein receptor (SNARE) proteins [89,90]. To date, less is known about how autophagosomes fuse with endosomes to form amphisomes. The accumulation of intracellular calcium is necessary for the fusion of autophagosomes with MVBs [95,96,97,98]. The inhibition of autophagosome fusion with the endosome by overexpressing the dominant negative mutant of the AAA ATPase, suppressor of K^+^ transport growth defect 1 (SKD1), suggests that SKD1-dependent membrane trafficking of endosomes is required for the formation of amphisomes [99]. The decoration of Rab11 on MVBs and the activation of Rab11, rather than Rab7, are needed for the fusion of autophagosomes with MVBs [98,100]. In addition to Rab11, vesicle-associated membrane protein 3 (VAMP3) is also necessary for fusion between MVBs and autophagosomes [101]. The ESCRT components, including VPS25, VPS32, VPS28, Deep Orange, and TSG101, function to regulate autophagosome fusion to endosomes [84,102,103,104,105,106]. ATG9 governs the formation of intraluminal vesicles in amphisomes [107], and IκB kinase (IKK) promotes amphisome formation by phosphorylating soluble NSF attachment protein (SNAP) 23 [108]. The zinc finger FYVE-type containing 26 (ZFYVE26, also named Spastizin) can interact with Rab5 and Rab11, and the hereditary spastic paraparesis-related mutation of ZFYVE26 leads to a defect in the fusion between autophagosomes and endosomes, implying that ZFYVE26 may be involved in amphisome formation. A deficiency in Niemann-Pick type C1 (NPC1), a transmembrane NPC1 protein shown to mediate cholesterol efflux from late endosomes, impairs the fusion of autophagosomes with endosomes by interfering with the recruitment of SNARE machinery to endosomes, suggesting that NPC1 plays a role in the formation of amphisomes [109,110]. Microtubules and the dynein–SNAPIN motor–adaptor complex function in autophagosome movement for fusion with endosomes [111,112,113]. The induced formation of homotypic fusion and protein sorting (HOPS) bodies, which are characteristically similar to amphisomes in cells lacking the HOPS complex, implies that the HOPS complex is unnecessary for the fusion of autophagosomes with endosomes [114].

Mature autophagosomes can also directly fuse with lysosomes, forming autolysosomes, in which lysosomal hydrolases degrade the sequestered contents (Figure 1) [20,21,22,23,24,56,94]. Several functional molecules, including the Rab family of small GTPases [115,116,117,118,119], SNARE proteins [120,121], tethering factors [122,123,124], and phosphoinositides (Figure 2) [118,125,126,127], have been shown to participate in the membrane fusion of autophagosomes with lysosomes. In addition, the movement of autophagosomes and lysosomes through the cytoskeleton, including microtubules and microfilaments, also facilitates autophagosome–lysosome fusion [115,116,117,118,119,128,129,130,131]. At the final stage of autophagy, the activation of mTOR by newly supplied nutrients inhibits autophagy initiation and promotes autophagic lysosome reformation (ALR), leading to autophagy termination [132]. Many cellular proteins, such as clathrin-mediated endocytosis proteins, microtubule motor proteins, kinesin, the ubiquitin E3 ligase Cullin 3-Kelch-like protein 20 (KLHL20), and lysosomal efflux permeases, have been shown to regulate the ALR to terminate autophagy [133,134,135,136,137,138,139,140,141,142,143]. Despite serving as a bulk and nonselective degradation process, autophagy can selectively eliminate specific cargoes, such as organelles and proteins; this process is referred to as “selective autophagy” (Figure 1) [144,145,146,147,148]. Various types of cargo receptors for selective autophagy have been shown to engulf degradative cargoes by recognizing proteins via ubiquitination and binding to adaptor proteins (Figure 1) [144,145,146,147,148]. By binding to ATG8/LC3 family proteins (which include the microtubule-associated protein 1A/1B-light chain 3 [LC3] and gamma-aminobutyric acid receptor-associated protein [GABARAP] subfamilies) on the membrane of autophagosomes via LC3-interacting regions (LIRs), ATG8-interacting motifs (AIMs), and GABARAP-interacting motifs (GIMs), cargo receptors associated with cargoes can be specifically delivered to autophagic vacuoles for degradation [144,145,146,147,148]. Various kinds of cargo receptors, including SQSTM1, the neighbor of BRCA1 (NBR1), calcium-binding and coiled-coil domain-containing protein 2 (Calcoco2, also named NDP52), optineurin (OPTN), BCL2/adenovirus E1B 19 kDa protein-interacting protein 3 (BNIP3)-like (BNIP3L/Nix), and Tax1-binding protein 1 (Tax1bp1), have been found to act as cargo receptors of selective autophagy [147,148,149,150,151,152].

## 3. The Regulatory Mechanism of Functional Molecules in Autophagosome–Lysosome Fusion

### 3.1. The Rab Family of Small GTPases

The Rab family of small GTPases fundamentally regulates the trafficking of intracellular membranes in eukaryotes by recruiting specific adaptor and motor proteins that are responsible for the transport and biogenesis of membranous organelles and vesicles [153,154,155,156,157]. In addition, the small GTPases of the Rab family also orchestrate tethering proteins for membrane fusion at endomembranes between organelles and vesicles [153,154,155,156,157]. The posttranslational isoprenylation of Rab family proteins and binding to specific guanine nucleotide exchange factors (GEFs) on the membrane regulate the targeting of Rab family proteins to distinct membranous compartments [153,154,155,156,157]; for example, Rab7 is associated with late endosomes and lysosomes. Through activation by GEF, membrane-associated Rab family proteins become GTP-bound and conformationally altered, allowing them to interact with effector proteins [153,154,155,156,157]. Reciprocally, GTPase-associated proteins (GAPs) drive the hydrolysis of GTP to GDP on Rab family proteins, inactivating and releasing proteins from the target membrane to the cytosol [153,154,155,156,157].

There are approximately 65 Rab family proteins in mammalian cells, and some of them have been shown to participate in autophagy regulation, particularly the fusion of autophagosomes with lysosomes. In the early 2000s, Gutierrez et al. first showed that Rab7 is necessary for autophagy in mammalian cells (Table 1) [116]. In that study, the authors demonstrated that the wild-type (WT) and dominant active forms (Q67L, GTP-bound) of Rab7 are associated with autophagic vacuoles in amino acid-starved CHO cells and that the inactive form (T22N, GDP-bound) of Rab7 is not [116]. Moreover, the overexpression of Rab7 T22N in amino acid-deprived and rapamycin-treated cells increases the size of autophagosomes labeled with an LC3 antibody [116]. These results suggest that Rab7 may regulate the formation of autophagic vacuoles at the late stage of autophagy, presumably through its GTPase activity [116]. Jager and colleagues subsequently reported that Rab7 is enriched in biochemically fractionated autophagic vacuoles in the rat liver [115]. By immunoelectron microscopy, the authors showed that Rab7 could be detected within initial-stage autophagic vacuoles (AVis), which contain intact cytoplasm, and late-stage/degradative autophagic vacuoles (AVds), in which lysosome-associated protein 1 (LAMP1) is localized in amino acid-deprived HeLa cells [115]. Moreover, gene knockdown of Rab7 and expression of Rab7 T22N dramatically decrease the number of LC3- and LAMP1-positive AVds in amino acid-starved HeLa cells [115]. These studies imply that Rab7 is required for the maturation of autophagic vacuoles at the late stage of autophagy [115].

Liang et al. reported that UVRAG, a Beclin 1-binding protein that induces autophagosome formation by activating PI3KC3 complex activity [167], can facilitate the transport of autophagosomes to late endosomes for autolysosome maturation in HeLa cells by increasing Rab7 GTPase activity (Table 1) [67], enhancing the colocalization of Rab7 and LC3-positive autophagosomes [67], and promoting the recruitment of the Rab-interacting lysosomal protein [67], a downstream effector of Rab7 required for endocytic transport [168]. Recent studies have indicated that the stacking protein GORASP2/GRASP55 in the Golgi apparatus and the mitochondrial protein Miga positively promote the fusion of autophagosomes and lysosomes through interactions with UVRAG [169,170]. Like in UVRAG, the Rab7-interacting protein pleckstrin homology domain-containing family M member 1 (PLEKHM1) was also reported to promote autophagosome–lysosome fusion in HeLa cells by binding to the HOPS complex and ATG8/LC3 family proteins [123,158] (Table 1). Baba and colleagues showed that endosomal phosphatidylinositol 4,5-bisphosphate (PtdIns(4,5)P) promotes the conversion of GTP-bound Rab7 to GDP-bound Rab7, leading to Rab7 release from PLEKHM1 and repressing autophagosome–lysosome fusion [159]. Very recently, Heo et al. demonstrated that tripartite motif (TRIM) 22, a ubiquitin E3 ligase, can enhance the association between PLEKHM1 and GABARAP family proteins to facilitate the fusion of autophagosomes with lysosomes [171]. In contrast to UVRAG and PLEKHM1, another Rab7-interacting protein, RUN domain and cysteine-rich domain-containing Beclin 1-interacting protein (Rubicon), was shown to suppress the fusion of autophagosomes with lysosomes (Table 1) [32,68,158]. A recent study demonstrated that nuclear receptor binding factor 2 (NRBF2), a component of the PI3KC3 complex [172], may activate Rab7 by interacting with the Rab7 GEF complex CCZ1-MONA1, thus positively regulating autolysosome maturation (Table 1) [160]. In this study, the authors found that gene knockout of NRBF2 in mouse neuroblastoma N2a cells and in the brain tissues of mice led to significant increases in LC3-II and SQSTM1 expression [160]. Additionally, NRBF2 can specifically localize within autolysosomes in N2A cells treated with Torin1, an mTOR inhibitor capable of inducing autophagy [160]. Moreover, NRBF2 was shown to interact with the CCZ1-MONA1 GEF complex via its MIT domain and bridge the association between PI3KC3 and the CCZ1-MONA1 complex, thereby promoting the Rab7-GEF interaction and Rab7 activation via the newly generated PtdIns(3)P [160]. These results indicate that NRBF2 is a Rab7 effector required for the maturation of autolysosomes [160].

In addition to Rab7, Rab2, an essential regulator of vesicle trafficking between the ER and Golgi apparatus, has been shown to regulate autophagosome–lysosome fusion [112,161,162,163,173]. The GTP-bound form of Rab2 (Rab2^GTP^) was demonstrated to directly bind to the HOPS complex component, VPS39 [161,174,175]. Gene silencing of Rab2 in *Drosophila melanogaster* leads to the accumulation of autophagosomes and amphisomes [176] and represses the formation of degradative autolysosomes [161]. In contrast, Rab2^GTP^ predominantly colocalizes with Rab7 and increases the formation of degradative autolysosomes in starved cells (Table 1) [161]. Similarly, gene knockdown of Rab2A in MDA-MB-231 cells results in the accumulation of LC3 within LAMP1-positive amphisomes [161], suggesting that Rab2 may participate in the fusion of autophagosomes with lysosomes in mammalian cells. Soon after, Ding et al. reported that autophagy activation by Torin1, an mTOR inhibitor, in U2OS cells leads to the translocation of Rab2 from the Golgi apparatus to autophagic vacuoles (Table 1) [162]. Gene knockout of Rab2 in U2OS cells suppresses the Torin1-induced formation of endogenous LC3-, ATG16L-, and ULK1-positive puncta [162]. In addition, Rab2 can interact with ULK1 and ATG13, and Rab2 overexpression promotes the phosphorylations of ULK1 at serine (Ser) 555, of ATG9 at Ser14, and of ATG14 at Ser29; in addition, Rab2 gene knockout diminishes the phosphorylation of these proteins, suggesting that Rab2 is required for autophagy initiation [162]. Moreover, the direct binding of Rab2 to the Rubicon-like autophagy enhancer (RUBUNL) and syntaxin 17 (Stx17) leads to the formation of an autophagosomal trimeric complex that recruits the HOPS complex for autophagosome–lysosome fusion [162]. Gene knockout of Rab2 represses the formation of mCherry^+^/GFP^−^ puncta of mCherry-GFP-LC3 (referred to as autolysosomes) in Torin1-treated U2OS cells [162]. Consistently, Rab2 knockout inhibits the colocalization of endogenous LC3 with LAMP1, but Rab2 overexpression promotes the colocalization of these proteins [162]. These results indicate that Rab2 crucially regulates autophagosome–lysosome fusion in mammalian cells.

Recently, Zhong’s group further confirmed that Rab2 specifically interacts with VPS39 but not VPS41, while Rab7 can bind VPS39 or VPS41 in HEK293 cells [112,163]. Rab2 and Rab7 coordinately promote lipid mixing between proteoliposomes containing the Stx17-associated SNARE complex, but the addition of the HOPS complex fails to further increase the membrane fusion of reconstituted proteoliposomes, suggesting that the pairing of Rab2 on autophagosomes and Rab7 on lysosomes cannot appropriately recruit the HOPS complex [112,163]. The authors further revealed that another Rab family of small GTPases, Rab39A, can directly bind to Stx17 and the HOPS complex components, VPS39 and VPS41 (Table 1) [112,163]. Autophagy induction by Torin1 induces the colocalization of Rab39A with autophagosomes and autolysosomes in U2OS cells [112,163]. Gene knockout of Rab39A inhibits autophagic flux in Torin1-treated 293T cells and induces the accumulation of endogenous LC3 puncta in U2OS cells [112,163]. Overexpression of Rab39A enhances the interaction of Stx17 with VPS33A and promotes the binding of Rab2 to VPS39, but prevents the interaction of Rab7 with VPS39 and VPS41 [112,163], suggesting that Rab39A may facilitate the recruitment of the HOPS complex. In addition, Rab2 and Rab39A promote the assembly of the HOPS complex to facilitate SNARE-mediated membrane fusion between proteoliposomes [112,163]. Moreover, C-terminal prenylation and GTP loading are required for the targeting of Rab39A to membranes, recruitment of the HOPS complex to autophagic vacuoles, and membrane fusion of proteoliposomes [112,163]. Furthermore, gene depletion of C9orf72, a GEF of Rab39A, leads to the accumulation of LC3^+^ puncta, the inhibition of autophagic flux, and a defect in HOPS complex recruitment, suggesting the pivotal role of C9orf72 activation of Rab39A in regulating autophagosome–lysosome fusion [112,163]. Notably, a recent study showed that gene silencing of Rab2 and Rab7 in *Drosophila* adult brains leads to the accumulation of ubiquitin-positive structures and mCherry-Atg8a-positive puncta [173]. Gene knockdown of Rab2, Rab7, and Arf-like GTPase 8 (Arl8) in *Drosophila* shortens lifespan [173]. In contrast, the activation of Rab2 and Arl8, but not Rab7, via the overexpression of their catalytically active (CA) (GTP-bound) forms extends lifespan [173]. Moreover, overexpression of Rab2-CA and Arl8-CA in fly neurons in the Parkinson’s disease model (harboring an A53T mutated form of α-synuclein) activates autophagy, promotes the clearance of A53Tα-Synuclein, and improves cell survival [173]. These studies imply the physiological significance of Rab2 and Arl8-regulated autophagy in longevity in vivo.

Rab33B and Rab37 were shown to promote autophagosome maturation by interacting with ATG16 and ATG5, respectively (Table 1) [177,178,179]. The GAP of Rab33B, ornithine aminotransferase-like 1 (OATL1, also known as TBC1D25), was shown to directly bind to ATG8/LC3 homologs (Table 1) [164]. OATL1 overexpression delays the fusion of autophagosomes with lysosomes through its ATG8/LC3-binding ability and GAP activity. Additionally, OATL1 overexpression suppresses Rab33B activation [164]. The activation of Rab33B by the overexpression of a GTPase-deficient mutant of Rab33B (QL) represses the fusion between autophagosomes and lysosomes [164]. Rab24 and Rab21 also facilitate autophagosome–lysosome fusion by interacting with the Rab-interacting lysosomal protein (RILP) and the endolysosomal trafficking of VAMP8 (Table 1) [165,166,180]. Unlike the Rab family of small GTPases that function in autophagosome–lysosome fusion, autolysosomal Rab32 and Rab38 were recently shown to promote the retrieval of Stx17 from autolysosomes through the autophagosomal components recycling (ACR) complex, which comprises sorting nexin (SNX) 4, SNX5, and SNX17 [181]. The biogenesis of the lysosome-related organelle complex-3 (BLOC-3) complex, which contains GEFs, Hermansky–Pudlak syndrome (HPS) 1 and HPS4, activates Rab32 and Rab38 to promote Stx17 retrieval from autolysosomes [181]. In contrast, Tre2/Bub2/Cdc16 domain-containing protein (RUTBC1), a GAP of Rab32 and Rab38, suppresses the retrieval of Stx17 from autolysosomes [181].

These studies collectively imply that the Rab family of small GTPases may promote autophagosome–lysosome fusion by mediating the recruitment of functional molecules to the vicinity of the fusion site and facilitating autophagosome movement toward the lysosome (Figure 3). Thus far, how these Rab family members of small GTPases translocate to autophagosomes has not been determined. Although the formation of amphisomes by the fusion of autophagosomes with late endosomes containing Rab7 may help autophagosomes acquire Rab7 for further fusion with lysosomes, it remains unclear how upstream signaling triggers the autophagosomal translocation of these Rab family proteins. In addition, whether other Rab family members of small GTPases compensate for autophagosome–lysosome fusion is still unknown. Further investigations are needed to unveil the molecular process by which Rab family protein translocation to autophagosomes is induced and to elucidate the detailed role(s) of each Rab family of GTPases in autophagosome–lysosome fusion.

### 3.2. SNAREs

SNAREs are a family of eukaryotic proteins responsible for the fusion of intracellular membranes [148,157,182,183]. Over sixty SNAREs have been identified in mammalian cells, each of which contains a C-terminal transmembrane domain (TMD) for anchoring to the membrane, a conserved SNARE motif, and a variable N-terminal domain harboring antiparallel helix bundles [148,157,182,183]. According to their association with donor and acceptor compartments, SNAREs are classified into vesicle membrane-SNAREs (v-SNAREs) and target membrane-SNAREs (t-SNAREs). Monomeric SNAREs can assemble into a heterooligomeric and helical core complex of four intertwined proteins by associating SNARE motifs with helical bundles [148,157,182,183]. There are sixteen layers of interacting highly hydrophobic side chains in the central bundle of this core complex [148,157,182,183]. In addition, three glutamine (Q) residues and one arginine (R) are highly conserved in the SNARE motif, which is located in the central “0” layer [148,157,182,183]. The four parallel helical bundles comprise three Q-SNAREs (Qa, Qb, and Qc) and one R-SNARE between the fused membranes; thus, this four-helical core complex is often called the QabcR complex [148,157,182,183]. The assembly of the QabcR trans complex is driven by two accessory proteins, namely, Sec1/Munc-18 (SM) proteins, complexes associated with tethering containing helical rods (CATCHR) proteins, and multisubunit tethering complexes (MTCs) [148,157,182,183]. After the trans-complex is formed, SNAREs are zippered toward the membrane-anchoring domain on the membrane (“power stroke”, which converts the trans-complex from a loose state to a tight state), thus allowing membrane fusion, in which this trans-complex becomes a cis-complex [148,157,182,183]. The disassembly of the SNARE cis-complex is mediated by the N-ethylmaleimide-sensitive factor (NSF) protein and its cofactor, SNAP [148,157,182,183].

In the late 2000s, Colombo’s group first reported that vesicle-associated membrane protein 7 (VAMP7) is a functional V-SNARE involved in the fusion of autophagosomes and lysosomes (Table 2) [101]. The authors found that starvation of human leukemic K562 cells triggers the colocalization of VAMP7 with RFP-LC3-labeled autophagosomes as well as dye-quenched bovine serum albumin (DQ-BSA) and monodansylcadaverine (MDC)-labeled autolysosomes [101]. Moreover, the overexpression of the N-terminal domain of VAMP7, which lacks the R-SNARE motif and the TMD, decreases the number of autolysosomes [101], suggesting that the functionality of VAMP7 is required for autophagosome–lysosome fusion. Furuta et al. subsequently showed that in addition to VAMP7, vesicle transport through interaction with t-SNARE homolog 1B (Vti1b) and VAMP8 colocalize with GFP-LC3-labeled autophagosomes in HeLa cells infected with group A Streptococcus (GAS) (Table 2) [184]. Further analysis revealed that Vti1b is localized mainly on the RFP^+^/GFP^+^ puncta of mRFP-GFP-LC3 (referred to as autophagosomes), whereas VAMP8 is retained on the RFP^+^/GFP^−^ puncta of mRFP-GFP-LC3 (referred to as autolysosomes) in starved HeLa cells [184]. Gene silencing of Vti1b and VAMP8 decreases the colocalization of RFP-LC3 with LAMP1 and increases the number of RFP^+^/GFP^+^-autophagosomes of the mRFP-GFP-LC3 reporter in nutrient-starved HeLa cells, supporting the positive roles of Vti1b and VAMP8 in regulating the fusion between autophagosomes and lysosomes [184].

In the early 2010s, Mizushima’s group reported that the ER-associated Qa-SNARE Stx17 is localized to LC3-positive autophagic vacuoles in nutrient-deprived mouse embryonic fibroblasts (MEFs) (Table 2) [185]. Gene knockdown of Stx17 leads to the accumulation of autophagosomes containing intact cytoplasm in nutrient-starved HeLa cells [185]. Moreover, Stx17 was shown to interact with VAMP8, an R-SNARE localized in late endosomes and lysosomes. Gene silencing of VAMP8 similarly increases the number of GFP-LC3-labeled puncta and decreases autophagic flux [185]. Moreover, the authors found that SNAP29, a Stx17-interacting Qbc-SNARE, can enhance the interaction between Stx17 and VAMP8 [185]. Similarly, SNAP29 knockdown blocks autophagic flux and induces the accumulation of GFP-LC3 puncta [185]. Most importantly, Stx17 was shown to localize to autophagosomes through glycine zipper-like motifs in the TMD [185]. These results suggest that Stx17 is an autophagosomal SNARE that regulates autophagosome–lysosome fusion by interacting with SNAP29 and lysosomal VAMP8 [185]. Huang et al. showed that mTORC1 can suppress the formation of the Stx17-SNAP29-VAMP8 complex by phosphorylating VAMP8, thus interfering with the fusion between autophagosomes and lysosomes (Table 2) [186]. Sec1 family domain-containing protein 1 (SCFD1) binds to Stx17 and VAMP8 of the SNARE complex [186]. The mTORC1-catalyzed phosphorylation of VAMP8 inhibits the interaction between SCFD1 and the SNARE complex, and the gene knockdown of SCFD1 blocks the fusion of autophagosomes to lysosomes [186]. Very recently, Huang and colleagues further demonstrated that SCFD1 is acetylated at lysine (Lys) residues, i.e., Lys126 and Lys515, by lysine acetyltransferase 2B (KAT2B/PCAF) and deacetylated by sirtuin 4 (SIRT4) (Table 2) [187]. The deacetylation of SCFD1 by nutrient deprivation is driven by its phosphorylation at Ser303 and Ser316 by 5′AMP-activated protein kinase (AMPK) [187]. Additionally, the SIRT4-mediated deacetylation of SCFD1 promotes the assembly of the Stx17-SNAP29-VAMP8 complex to facilitate autophagosome–lysosome fusion [187]. These findings indicate the functional role of SCFD1 acetylation and deacetylation in the formation of the SNARE complex involved in autophagosome–lysosome fusion [187]. Furthermore, Yoshimori’s group demonstrated that Stx17 participates in autophagy initiation by promoting IM/phagophore formation at the ER-mitochondria contact site [50]. The TANK-binding kinase 1 (TBK1)-mediated phosphorylation of Stx17 at Ser202 was shown to promote the translocation of Stx17 from the Golgi apparatus to the cytosolic preautophagosomal structure (PAS) to initiate autophagy [197].

Soon afterward, Jiang et al. showed that Stx17 interacts with the HOPS tethering complex, which contains VPS11, VPS16, VPS18, VPS33, VPS39, and VPS41 (Table 2) [122]. Individual gene knockdown of VPS33A, VPS16, and VPS39 in starved HeLa cells inhibited autophagic flux [122]. Gene silencing of VPS33A increased the number of RFP^+^/GFP^+^ autophagosomes of the mRFP-GFP-LC3 reporter, similar to what was observed in Stx17-silenced cells. VPS33A or VPS29 knockdown leads to the accumulation of LC3- and Stx17-positive puncta in HeLa cells [122]. These studies imply that the HOPS complex functions in the fusion of autophagosomes with lysosomes by interacting with Stx17 [122]. Along with these studies, Saleed and colleagues used high-resolution fluorescence lifetime imaging (FLIM)-fluorescence resonance energy transfer (FRET) analysis to demonstrate that Stx17, SNAP29, and VAMP7, rather than VAMP8, form a heteromeric complex on autophagosomes in rapamycin-treated HeLa cells (Table 2) [188]. The phosphorylation of Stx17 at Ser2 was shown to inhibit its interaction with SNAP29 and VAMP7 [188]. Moreover, the authors showed that VPS33A knockdown leads to an increase in Stx17-positive puncta and interferes with the interaction of VAMP7 with Stx17 and SNAP29 [188], supporting the hypothesis that VPS33A plays a critical role in the assembly of the SNARE bundle for membrane fusion. Furthermore, VPS33A was shown to preferentially bind to phosphorylated and dephosphorylated Stx17 at the prefusion and fusion stages, respectively [188]. These results suggest that the interaction of VPS33A with Stx17, presumably through Ser2-mediated phosphorylation of Stx17, functions as a switch for the progression of prefusion to fusion in autophagosome–lysosome fusion [188]. In addition to phosphorylation, Stx17 is regulated by acetylation at Lys219 and Lys223 by the histone acetyltransferase CREB-binding protein (CBP) and deacetylation by histone deacetylase 2 (HDAC2) (Table 2) [189]. Shen and colleagues showed that upon nutrient-related autophagy activation, Stx17 is deacetylated, promoting the interaction of Stx17 with SNAP29 for assembly of the Stx17-SNAP29-VAMP8 complex and enhancing its binding to the HOPS complex for autophagosome–lysosome fusion [189]. These studies support the regulatory role of Stx17 acetylation/deacetylation in the fusion of autophagosomes with lysosomes [189]. A recent study reported that the stimulator of interferon genes (STING) directly binds to Stx17 via its C-terminal domain [190]. Also, the interaction of STING with Stx17 interferes with the assembly of the Stx17-SNAP29-VAMP8 complex and blocks this SNARE complex-mediated membrane fusion of proteoliposomes [190]. In addition, enhanced polymerization by STING phosphorylation disrupts the interaction between STING and Stx17 and relieves the inhibitory effect of STING on the assembly of the Stx17-SNAP29-VAMP8 complex, suggesting that STING activation promotes autophagosome–lysosome fusion [190]. Moreover, trafficking degradation-deficient mutants and lupus-associated STING mutations were shown to disrupt Stx17-SNAP29-VAMP8 complex assembly and inhibit autophagic flux [190].

Zhong’s group demonstrated that ATG14, a component of the PI3KC3 complex required for autophagy initiation, can directly bind to Stx17 and SNAP29 through the coiled-coil domain (CCD) of ATG14 and the SNARE motif of Stx17 (Table 2) [121]. Additionally, ATG14 was shown to colocalize with Stx17 on LC3-positive autophagosomes and LAMP1-positive autolysosomes in human osteosarcoma U2OS cells [121]. By performing an in vitro tethering assay using a single vesicle/liposome, the authors found that the recombinant ATG14 protein alone sufficiently promoted membrane tethering rather than lipid mixing of protein-free liposome membranes [121]. Moreover, purified ATG14 enhances lipid mixing and content mixing in proteoliposomes containing purified Stx17, SNAP29, and VAMP8 [121]. Deletion of the Stx17-interacting CCD within ATG14 dramatically diminishes its activation of membrane fusion [121]. Furthermore, ATG14 was shown to oligomerize homologously through four conserved cysteine (Cys) residues in its N-terminal domain [121]. Interference with ATG14 homo-oligomerization leads to an increase in autophagosomes in U2OS cells and reduces the ability of autophagosomes to promote vesicle tethering and lipid mixing [121]. These studies indicate that the Stx17-SNAP29-VAMP8 complex promotes the fusion of autophagosomes with lysosomes through ATG14 [121].

In contrast to the positive regulatory effect on autophagosome–lysosome fusion, Guo and colleagues demonstrated that the O-GlcNAcylation of SNAP29 by O-linked β-N-acetylglucosamine (O-GlcNAc) transferase (OGT) negatively regulates autophagosome–lysosome fusion (Table 2) [191]. Gene silencing of OGT was shown to increase the number of autophagosomes and autolysosomes in HeLa cells deprived of nutrients [191]. Additionally, OGT knockdown promoted the assembly of the Stx17-SNAP29-VAMP8 complex [191]. Moreover, the authors demonstrated that Ser2, Ser61, threonine (Thr) 130, and Ser153 of SNAP29 could be O-GlcNAcylated and that SNAP29 O-GlcNAcylation diminished the interaction between Stx17 and VAMP8 [191]. The loss of SNAP29 O-GlcNAcylation by mutation of these residues leads to the accumulation of LC3- and Stx17-positive puncta in HeLa cells [191]. Furthermore, the authors found that glucose deprivation and nutrient starvation decrease the O-GlcNAcylation of SNAP29 [191]. Recently, Pellegrini and colleagues reported that a stabilizer of the interaction between microtubules and the kinetochore complex, the cytotoxic small molecule (SM15), can induce SNAP29 O-GlcNAcylation to suppress the formation of the SNARE complex and block autophagic flux, thereby promoting reactive oxygen species (ROS)-induced cell apoptosis [192].

Like Stx17, another v-SNARE, YKT6, was shown to be localized on autophagosomes and to regulate the fusion of autophagosomes with lysosomes (Table 2) [193,194]. Matsui et al. reported that gene knockout of Stx17 (Stx17 KO) unexpectedly leads to a partial reduction in autophagosome–lysosome fusion in starved HeLa cells [194]. Gene silencing of YKT6 decreases the number of autolysosomes in starved WT cells and completely inhibits the formation of autolysosomes in starved Stx17 KO cells [194]. Moreover, the authors showed that YKT6 can form a SNARE complex with SNAP29 and syntaxin 7 (Stx7) and that YKT6 is recruited to autophagosomes through its N-terminal longin domain [194]. Later, in an in vitro fusion assay, Gao and colleagues further demonstrated that YKT6 is an autophagosome v-SNARE that mediates autophagosome–vacuole fusion [193]. These studies indicate that YKT6 and Stx17 independently regulate autophagosome–lysosome fusion [193,194]. A recent study showed that ULK1 can phosphorylate YKT6 at Thr156 [195]. Enhancement of ULK1-mediated YKT6 phosphorylation and the phosphor-mimetic mutant (T156E) of YKT6 inhibit the interaction between SNAP29 and YTK6, thus repressing the fusion of autophagosomes to lysosomes and blocking autophagic flux [195]. These studies suggest that ULK1 may phosphorylate YKT6 on autophagosomes to prevent premature fusion with lysosomes at the early stage of autophagy [195].

Very recently, Rong’s group demonstrated that the Stx17-SNAP47-VAMP7/VAMP8 complex regulates autophagosome–lysosome fusion in mitophagy, a specific form of selective autophagy for the turnover of mitochondria (Table 2) [196]. The authors showed that SNAP47 specifically localizes to mitophagosomes, and gene knockout of SNAP47 inhibits oligomycin/adriamycin (OA)- and hypoxia-induced mitophagic degradation [196]. In addition, SNAP47 also regulates other types of selective autophagy, including aggrephagy and ER-phagy [196]. SNAP47 can interact with Stx17 and VAMP7/VAMP8 to form a ternary SNARE complex, which functions in mitophagy activation and in vivo membrane fusion of proteoliposomes [196]. SNAP47 is recruited to autophagosomes through its binding to PtdIns(4, 5)P2 via the PH domain and interaction with ATG8/LC3s via LIR motifs [196]. SNAP29 is not involved in autophagosome–lysosome fusion, presumably due to the unchanged O-GlcNAcylation of SNAP29 during mitophagy [196]. The inhibited degradation of LC3-II and p62/SQSTM1 in starved SNAP47 knockdown cells and the lack of further suppression of GFP acidification of mRFP-GFP-LC3 by gene silencing of SNAP29 in SNAP47 knockdown cells suggest that SNAP47 and SNAP29 could redundantly regulate autophagosome–lysosome fusion [196].

These studies indicate that two SNARE complexes, the Stx17-SNAP29-VAMP7/8 complex and the YKT6-SNAP29-Stx7 complex, promote autophagosome–lysosome fusion by bridging two opposite membranes (Figure 3). So far, whether these two complexes are functional independently or cooperatively is still unclear. A recent study showed that YKT6 can form a priming complex with Stx17 and SNAP29 on autophagosomes, after which lysosomal VAMP8 displaces YKT6 to form the Stx17-SNAP29-VAMP8 complex for membrane fusion between autophagosomes and lysosomes [198]. The released YKT6 subsequently interacts with SNAP29 and Stx7 to assemble the YKT6-SNAP29-Stx7 complex, which also facilitates membrane fusion [198]. Therefore, this study suggests that these two SNARE complexes may collaborate in the regulation of autophagosome–lysosome fusion. However, further studies are needed to understand whether and how these two SNARE complexes function differently in each type of selective autophagy and in various mammalian tissues. Additionally, the timely regulation of Stx17 and YKT6 translocation to autophagosomes at the late stage of autophagy needs further investigation.

### 3.3. Tethering Factors

To facilitate membrane fusion, tethering proteins are recruited to bridge the association between opposite membranes and trigger the assembly of the SNARE complex [199,200,201,202]. Three classes of tethering proteins have been identified, namely, CATCHRs, Class C MCTs, and long coiled-coil proteins. The heteromeric HOPS complex belongs to the class C MCTs and is composed of six different subunits: VPS33A, VPS16, VPS11, VPS18, VPS39, and VPS41. Mizushima’s group first demonstrated that the HOPS complex regulates the fusion of autophagosomes with lysosomes by interacting with Stx17 (Table 3) [122]. Among these six subunits, VPS33A can differentially bind to Stx17 at the prefusion and fusion stages via Ser2 phosphorylation of Stx17, suggesting that the HOPS complex controls the progression of membrane fusion between autophagosomes and lysosomes [188]. In addition to VPS33A binding to Stx17, VPS39 and VPS41 have been shown to interact with PLEKHM1, which can bind to ATG8/LC3 on autophagosomes and to Rab7 on late endosomes and lysosomes, indicating that the HOPS complex is recruited to the membrane fusion site by PLEKHM1, thus bridging the fusion between autophagosomes and lysosomes (Table 3) [123,158]. Consistent with this, ATG8/LC3 family proteins were shown to be essential for autophagosome–lysosome fusion rather than autophagosome maturation [203]. Among these proteins, GABARAP promotes the recruitment of PLEKHM1 for autophagosome fusion with lysosomes [203]. In addition to PLEKHM1, the protein associated with UVRAG, an autophagy enhancer (Pacer) that acts as a stimulator of the PI3KC3 complex by antagonizing Rubicon, was reported to direct the formation of the HOPS and PI3KC3 complexes on autophagosomes via Stx17 and phosphoinositides (Table 3) [204]. Very recently, Rab39A was reported to trigger the assembly of the HOPS complex, and the C9orf72, GEF-mediated activation of Rab39A was shown to promote the recruitment of the HOPS complex to autophagosomes, thus facilitating autophagosome–lysosome fusion (Table 3) [112]. ATG14 also serves as a tethering factor for autophagosome–lysosome fusion by interacting with Stx17 and SNAP29 of the SNARE complex via its CCD (Table 3) [121]. In addition, ATG14 itself harbors the ability to tether two liposomal membranes together, and ATG14 promotes the mixing of lipids and the contents of proteoliposomes reconstituted with Stx17, SNAP29, and VAMP8 through its CCD and homo-oligomerization [121].

Ectopic P granules protein 5 (EPG5), a Rab7 effector, can function as a tether by interacting with Rab7 on late endosomes, lysosomes, and LC3 autophagosomes (Table 3) [126]. EPG5 also promotes the formation of the Stx17-SNAP29-VAMP8 SNARE complex and stabilizes this assembled complex [126]. In addition, EPG5 induces lipid mixing between the donor proteoliposome reconstituted with Stx17-SNAP29 and the Rab7- and VAMP7-containing acceptor proteoliposome [126]. These findings indicate that EPG5 acts as a tether for autophagosome fusion with lysosomes. A later study showed that EPG5 preferentially binds to GABARAP subfamily proteins, thus promoting the recruitment of mitochondria to sites of selective autophagy (Table 3) [208]. Golgi reassembly stacking protein 55 (GRASP55) is another tethering factor that interacts with LC3-II on autophagosomes and LAMP2 on lysosomes, thereby facilitating the fusion of autophagosomes and lysosomes (Table 3) [205]. Glucose deprivation leads to the de-O-GlcNAcylation of GRASP55, which results in its release from the Golgi apparatus to autophagosomes to mediate fusion with lysosomes [205].

Zhong’s group reported that tectonin beta-propeller repeat containing 1 (TECPR1) can interact with the ATG12-ATG5 conjugate, thus recruiting ATG12-ATG5 to autolysosomes (Table 3) [206]. ATG12-ATG5-bound TECRP1 can specifically bind to PtdIns(3)P to ensure its critical function in autophagosome maturation, suggesting that TECPR1 may act as a tether for autophagosome–lysosome fusion [206]. The BIR repeat containing ubiquitin-conjugating enzyme (BRUCE) is a member of the inhibitor of apoptosis (IAP) family of proteins and was shown to play a tethering role in autophagosome–lysosome fusion by interacting with the Stx17, GABARAP, and GABARAPL proteins (Table 3) [207].

Together, these studies imply that several tethering factors can mediate autophagosome–lysosome interactions by facilitating SNARE complex assembly and promoting membrane fusion (Figure 3). However, it remains largely unknown whether these membrane tethers are necessary or functional independently. Further investigations are required to elucidate the interplay between these autophagosome–lysosome fusion tethers. In addition, whether and how these tethering factors promote autophagosome–lysosome fusion in different types of mammalian tissues in vivo needs further study.

### 3.4. Phosphatidylinositol Phosphates (Phosphoinositides)

Phosphoinositides are intracellular phospholipids in the cytoplasmic leaflet of eukaryotic membranes that essentially regulate cell signaling, cell growth, and membrane trafficking [209,210,211]. Phosphatidylinositol (PtdIns), the precursor of phosphoinositides, comprises inositol and phosphatidic acid and is connected by a phosphate group. Phosphatidylinositol is generated in the ER and can be phosphorylated by various phosphoinositide kinases to different species of phosphoinositides. Conversely, several phosphatases can dephosphorylate these phosphoinositides, ultimately converting them to phosphatidylinositol. PtdIns is phosphorylated by the PI3KC3 complex [26,31,32,33], which produces PtdIns(3)P for the recruitment of the IM/phagophore regeneration factors DFCP1 and WIPIs for autophagy initiation [26,31,32,34,35]. In addition to initiating autophagy, PtdIns(3)P has been shown to be required for the tethering function of TECRP1 to autophagosomes and lysosomes (Table 4) [206]. TECRP1 can bind to PtdIns(3)P through its pleckstrin homology (PH) domain and interact with the ATG12-ATG5 complex.

In addition, 1-phosphatidylinositol 3-phosphate 5-kinase (PIKFYVE) phosphorylates PtdIns(3)P to produce phosphatidylinositol (3,5)-bisphosphate (PtdIns(3,5)P2), which is crucial for maintaining the functions of lysosomes [213]. In contrast, PtdIns(3,5)P2 can be converted to PtdIns(3)P by inositol polyphosphate-5-phosphatase E (INPP5E), the mutation of which has been shown to be linked to ciliopathies in Joubert syndrome [216]. Hasegawa et al. first reported that gene knockdown of INPP5E suppresses autophagic flux, induces the accumulation of LC3-positive autophagosomes, and inhibits the colocalization of LC3^+^ puncta with LAMP1 (Table 4) [212]. Additionally, the lysosomal localization and phosphatase activity of INPP5E were shown to regulate autophagy [212]. The authors further demonstrated that INPP5E promotes the conversion of lysosomal PtdIns(3,5)P2 to PtdIns(3)P and facilitates microfilament polymerization by activating cortactin, thus promoting the fusion of autophagosomes and lysosomes [212].

Phosphatidylinositol 4-phosphate (PtdIns(4)P) generated by phosphatidylinositol 4-kinase IIα (PI4KIIα) also positively regulates the fusion of autophagosomes and lysosomes (Table 4) [214]. Wang and colleagues reported that nutrient starvation induces the translocation of PI4KIIα and PtdIns(4)P from the trans-Golgi network (TGN) to LC3-positive autophagosomes in HeLa cells [214]. Gene silencing of PI4KIIα was shown to inhibit autophagosome–lysosome fusion and induce autophagosome accumulation. GABARAPs, rather than LC3 subfamily proteins, mediate the recruitment of PI4KIIα to autophagosomes [214]. The induced accumulation of GFP-LC3 autophagosomes in GABARAP-knockdown cells can be reduced by overexpressing WT but not the kinase-dead (KD) PI4KIIα [214]. These results revealed the functional roles of PI4KIIα and PtdIns(4)P in the fusion of autophagosomes with lysosomes [214]. Very recently, Laczkó-Dobos and colleagues demonstrated that Stx17 binds to PtdIns(4)P in vitro via its C-terminal positively charged amino acids and colocalizes with PtdIns(4)P on autophagosomes in starved HEK293 and U2OS cells [215]. Gene silencing of PI4KIIα and treatment with the PI4KIIα inhibitor, NC03, suppress the autophagosomal recruitment of Stx17 and inhibit autophagosome–lysosome fusion in starved HEK293 cells [215]. Substitution of the positively charged amino acids to alanine at the C-terminal region of Stx17 impairs its interaction with PtdIns(4)P and inhibits its autophagosomal localization [215].

In addition, another study demonstrated that phosphatidylinositol-4-phosphate 5-kinase gamma (PIP5Kγ) induces the conversion of PtdIns(4)P to phosphatidylinositol (4,5)-bisphosphate (PtdIns(4, 5)P2) in late endosomes, leading to the inactivation of Rab7 and the dissociation of PLEKHM1 from late endosomes and lysosomes (Table 4) [159]. These studies imply that PIP5K-catalyzed PtdIns(4, 5)P2 may promote the recycling of Rab7 and the release of PLEKHM1 from lysosomes after autophagosome–lysosome fusion [159]. Very recently, PtdIns(4, 5)P2 was shown to facilitate the recruitment of SNAP47 to autophagosomes to promote the assembly of the Stx17-SNAP47-VAMP7/VAMP8 complex, which is required for autophagosome–lysosome fusion during mitophagy [196].

These studies collectively suggest that the phosphoinositides PtdIns(3)P and PtdIns(4)P may promote membrane fusion between autophagosomes and lysosomes by recruiting tethering factors and promoting the polymerization of microfilaments (Figure 3). Oppositely, other phosphoinositides, PtdIns(3,5)P2 and PtdIns(4, 5)P2 can suppress autophagosome–lysosome fusion by inactivating Rab7 and dissociating PLEKHM1 from lysosomes. However, how these phosphoinositides regulate the fusion of autophagosomes to lysosomes is largely unknown and needs further investigation. Also, additional studies are required to understand how these phosphoinositides are translocated to the fusion site over time.

### 3.5. Cytoskeleton and Motor Proteins

Cytoskeletal structures, such as microtubules and associated motor proteins, are necessary for the intracellular transport of organelles, including autophagosomes [217,218,219]. The movement of autophagosomes from the cytosol to the perinuclear region, where late endosomes and lysosomes are primarily localized, is detrimental to autophagosome–lysosome fusion [220]. Yoshimori’s group first used time-lapse live-cell imaging to reveal the movement of GFP-LC3-labeled autophagosomes toward γ-tubulin-positive centrosomes in HeLa cells (Table 5) [221]. Microinjection of an anti-LC3 antibody was shown to interfere with the association of RFP-LC3 puncta and Alexa 488-BSA-positive late endosomes and lysosomes [221]. Additionally, the authors demonstrated that impairing the biological activity of dynein, which forms a minus-end-directed motor complex with dynactin, with an anti-dynactin antibody disrupts the movement of GFP-LC3-positive autophagosomes [221]. Similarly, the overexpression of dynamitin, which inhibits the dynein–dynactin-mediated minus-end transport of organelles, can repress the delivery of GFP-LC3^+^ autophagosomes [221]. These studies suggest that the dynein–dynactin motor complex and associated minus-end transport are essential for autophagosome movement and subsequent fusion with lysosomes [221]. Similarly, Khobrekar et al. demonstrated that RILP, a dynein–dynactin motor complex-interacting Rab7 effector [119,168], can interact with LC3 on autophagosomes via its LIR but is independent of its ability to bind to Rab7 (Table 5) [222]. Gene knockdown of RILP was shown to inhibit the movement of RFP-LC3-positive autophagosomes and GFP-Rab7-labeled late endosomes and lysosomes [222]. These studies imply that RILP not only mediates the delivery of late endosomes and lysosomes through Rab7 but also regulates the retrograde transport of autophagosomes by binding to LC3 [222]. Notably, Rab7-RILP was recently shown to facilitate the minus-end-directed movement of late endosomes on microtubules for fusion with lysosomes through the HOPS complex, which is recruited by Rab2 to late endosomes, and Arl8 and the BLOC one-related complex (BORC), which are located on lysosomes [223]. These studies imply that Rab7-RILP also participates in endosome-lysosome fusion by moving late endosomes toward lysosomes [223].

Wijddeven and colleagues reported that cholesterol may regulate the subcellular localization of autophagosomes via the oxysterol-binding protein (OSBP)-related protein 1L (ORP1L) (Table 5) [130], a cholesterol sensor that can bind to RILP and control late endosome positioning [224]. The authors found that the depletion of intracellular cholesterol by statins leads to the scattering of autophagosomes throughout the cytoplasm, whereas the induced accumulation of endosomal cholesterol by U1866A results in the perinuclear localization of autophagosomes [130]. A reduction in the perinuclear positioning of autophagosomes in cholesterol-depleted cells can be restored by knocking down ORP1L [130]. Further analysis revealed that ORP1L controls the RILP-mediated minus-end transport of autophagosomes and PLEKHM1 and HOPS complex recruitment to Rab7-positive late endosomes and lysosomes [130]. Moreover, RILP and PLEKHM1 assemble into a complex through the HOPS complex and promote their recruitment to late endosomes and lysosomes by Rab7, thereby facilitating autophagosome–lysosome fusion [130]. These results indicate that ORP1L and RILP cooperate in the transport of autophagosomes and in the fusion of autophagosomes and lysosomes [130].

In addition to the microtubule minus-end-directed movement of autophagosomes, Johansen’s group showed that another Rab7 effector, the novel FYVE and coiled-coil (CC) domain-containing protein (FYCO1), could bind to LC3B and Rab7, forming an adaptor complex to facilitate plus-end-directed autophagosome transport on microtubules (Table 5) [118]. The authors demonstrated that FYCO1 interacts with LC3B through its FYVE domain, a motif required for binding to PtdIns(3)P. FYCO1 was shown to be localized on the external membranes of autophagosomes, late endosomes, and lysosomes through its FYVE and dimerization coiled-coil (CC) domains, and that Rab7 binds to FYCO1 to promote the recruitment of FYCO1 to these membranes [118]. Moreover, FYCO1 was found to compete with RILP for binding to Rab7 and direct the plus-end and anterograde transport of autophagosomes through its kinesin-binding motif [118]. These results imply that FYCO1 functions by directing the plus-end-directed movement of autophagosomes toward the cell periphery [118]. A recent study indicated that nutrient starvation-induced phosphorylation of LC3B at threonine (T) 50 by serine/threonine-protein kinase 4 (STK4) prevents the FYCO1-directed anterograde movement of autophagosomes, thus redistributing autophagosomes to the perinuclear region [227].

Microfilament assembly and associated motors, such as myosin VI, also participate in the regulation of autophagosome–lysosome fusion. Yao’s group demonstrated that histone deacetylase-6 (HDAC6), a ubiquitin-binding deacetylase, controls the fusion of autophagosomes with lysosomes through F-actin assembly (Table 5) [131]. The authors showed that gene knockdown of HDAC6 interferes with the clearance of protein aggregates by selective autophagy but does not affect starvation-induced autophagy in MEFs [131]. Additionally, the induction of autophagosome accumulation was shown in HDAC6KO MEFs, and the purified autophagosomes and lysosomes from HDAC6KO MEFs were shown to efficiently fuse in an in vitro fusion assay [131]. Moreover, the F-actin network was shown to colocalize with protein aggregates through the HADC6-dependent recruitment of cortactin, a critical molecule of the F-actin polymerization machinery [131]. These studies imply that HDAC6 facilitates autophagosome–lysosome fusion by promoting cortactin-mediated assembly of microfilaments [131]. Soon afterward, Tumbarello and colleagues showed that gene silencing of myosin VI, an unconventional myosin used for microfilament minus-end-directed movement (Table 5) [228], results in the accumulation of autophagosomes and the disruption of the clearance of protein aggregates in MG132-treated HeLa cells [225]. Myosin VI was shown to be targeted to autophagosomes through its cargo-binding tail domain and interaction with NDP52, optineurin, and Traf6-binding protein (T6BP) [225]. By binding to the target of Myb1-like 2 (Tom1/Tom1L2), an endosomal sorting factor required for recognizing ubiquitinated cargoes, myosin VI is recruited to endosomes and subsequently mediates the delivery of the endosomal compartment to autophagosomes [225]. Moreover, gene silencing of Tom1 was shown to repress the fusion of autophagosomes to lysosomes [225]. These studies suggest that myosin VI regulates autophagosome–lysosome fusion by interacting with cargo receptors on autophagosomes and Tom1/Tom1L2 on the endosomal membrane [225]. Very recently, Son et al. further demonstrated that the intermediate filament protein keratin 8 (KRT8) interacts with microfilaments via plectin (PLEC), stabilizing actin filaments and thus promoting autophagosome–lysosome fusion (Table 5) [226].

These studies imply that transport through the cytoskeleton and motor system, predominantly by movement on the microtubules, is critical for encountering and fusion between autophagosomes and lysosomes (Figure 3). In addition to the microtubules, microfilament polymerization is likely functional for autophagosome–lysosome fusion. It remains unclear how these cytoskeletal motors collaborate to precisely regulate the movement of autophagosomes and lysosomes to the fusion site. Thus, further studies are needed to dissect the interconnection between the cytoskeleton and associated motor involved in concerted action to facilitate the transport of autophagosomes toward lysosomes for fusion.

### 3.6. Summary

Autophagosome–lysosome fusion is critical for completing the entire autophagy process, during which the sequestered material can be degraded. Several kinds of regulatory molecules act on this fusion process [19,20,23,56,229,230]. The Rab family of GTPases, particularly Rab7, on late endosomes and lysosomes plays a crucial role in recruiting tethering factors, such as PLEKHM1 and EPG5, for autophagosome–lysosome fusion (Figure 3). In addition, the Rab7 effector RILP mediates the transport of autophagosomes and lysosomes (Figure 2). Two major SNARE complexes, Stx17-SNAP29-VAMP8 and Stx17-SNAP29-VAMP7, have been shown to regulate the fusion of autophagosomes with lysosomes (Figure 3). In addition to transporting autophagosomes and lysosomes together, SNARE complexes promote the recruitment of membrane tethers, such as the HOPS complex and ATG14 (Figure 3). In addition to the Stx17-assembled SNARE complex, the YKT6-SNAP29-Stx7 SNARE complex also facilitates autophagosome–lysosome fusion and may compensate for the function of the Stx17 SNARE complex. The tethering factors, such as the HOPS complex, ATG14, EPG5, TECRP1, GRASP55, and BRUCE, are involved in membrane fusion between autophagosomes and lysosomes (Figure 3). These tethering proteins are recruited through their binding to the SNARE complex, the ATG12-ATG5 conjugate, and ATG8/LC3s on autophagosomes, and Rab7 and LAMP2 on lysosomes (Figure 3). In addition to proteins, phosphoinositides, including PtdIns(3)P and PtdIns(4)P, are associated with autophagosomes and may enhance autophagosome–lysosome fusion by recruiting TECRP1 and promoting microfilament polymerization, whereas PtdIns(4)P can be generated by PI4KIIα and translocated from the TGN to autophagosomes to promote fusion with lysosomes (Figure 3). In contrast, PtdIns(4, 5)P2 may antagonize the fusion of autophagosomes with lysosomes by inactivating Rab7 and dissociating PLEKHM1 from late endosomes and lysosomes. The minus-end-directed movement of autophagosomes through the dynein–dynactin complex is necessary for autophagosome–lysosome fusion (Figure 3). RILP and ORP1L mediate the interaction of the dynein–dynactin complex with autophagosomes by interacting with the Rab7 complex (Figure 3). The FYCO-Rab7-LC3 adaptor complex allows the plus-end-directed movement of autophagosomes to the periphery (Figure 3). Furthermore, HDAC6-associated microfilament polymerization and myosin VI also positively regulate autophagosome–lysosome fusion.

### 3.7. Deregulation of Autophagosome–Lysosome Fusion in Human Diseases

Interference with the formation of autolysosomes by disrupting the fusion of autophagosomes to lysosomes has been shown to be associated with the development of several human diseases, including neurodegenerative diseases, myopathies, sepsis, encephalopathy, and viral infection. Truncation of the EPG5 protein by recessive mutations in the human *EPG5* gene results in Vici syndrome, an early-onset neurodevelopmental disorder encompassing cardiomyopathy, hypopigmentation, and immunodeficiency [231,232]. The development and progression of Vici syndrome in human patients and mice are associated with the loss of functional EPG5 and disturbances in autophagosome–lysosome fusion and defective autophagy [126,231,232,233,234]. In addition to Vici syndrome, the reduced interaction of EPG5 with LC3 and the downregulation of Rab7 expression have been shown to impair autophagy at the late stage in platelets from human sepsis patients [235]. A defect in the fusion of autophagosomes with lysosomes induced by a VPS11 mutation is associated with leukoencephalopathy [236].

On the other hand, several human viruses subvert autophagosome–lysosome fusion to benefit the viral life cycle through accumulated autophagosomes. Hepatitis C virus (HCV) infection induces Rubicon expression and represses UVRAG expression at the early infection stage, thus transiently impairing the fusion of autophagosomes to lysosomes to increase autophagosome formation for viral RNA replication [237]. Coxsackievirus B3 (CVB3) interferes with Stx17-SNAP29-VAMP8 complex assembly by inducing the cleavage of SNAP29 and PLEKHM1 by CVB3 proteinase 3C, thereby disrupting the fusion between autophagosomes and lysosomes and preventing infected cells from undergoing cell apoptosis [238]. Similarly, another enterovirus, enterovirus D68 (EV-D68), can block autophagosome–lysosome fusion through protease 3C-mediated SNAP29 cleavage to enhance the production of infectious virions [239]. The influenza A virus (IAV) M2 protein can bind to TBC1D5/OATL1 to interrupt the interaction between Rab7 and TBC1D5/OATL1, repressing autophagosome fusion to the lysosome [240]. Severe acute respiratory syndrome coronavirus 2 (SARS-CoV-2) open reading frame 3a (ORF3a) was shown to interact with VPS39, thereby interfering with the assembly of the Stx17-SNAP29-VAMP7/VANP8 SANRE complex required for autophagosome–lysosome fusion [241,242]. Additionally, the SARS-CoV-2 ORF3a also targets UVRAG and interferes with the formation of the functional PI3KC3 complex for autophagosome fusion to lysosome [243]. This SARS-CoV-2 ORF3a-induced defect in autolysosome maturation promotes viral RNA replication and disturbs sphingolipid homeostasis in infected neuronal cells [243,244]. In addition to ORF3a, SARS-CoV-2 ORF7 can repress the fusion of autophagosomes to lysosomes by activating caspase 3-dependent cleavage of SNAP29 to facilitate virus replication [245]. These studies indicate that viruses can suppress autophagosome–lysosome fusion by targeting components of the fusion machinery, such as SNAP29 and UVRAG, to promote virus growth and viral-associated disease pathogenesis.

## 4. Conclusions and Perspectives

In recent decades, multiple mechanisms have been demonstrated to regulate autophagosome–lysosome fusion in mammalian cells. However, many questions remain unanswered. It remains unclear how the SNARE complex is recruited to the fusion site involved in autophagosome–lysosome fusion. Additionally, at least three SNARE complexes that function in the fusion of autophagosomes to lysosomes have been identified; however, how they differentially regulate this fusion step in different kinds of autophagy, such as selective autophagy, is unclear. Additionally, there are many tethers involved in autophagosome–lysosome fusion. Thus far, whether these proteins compensate for the functions of other proteins is unclear. Notably, how these tethering factors are precisely recruited for autophagosome–lysosome fusion is enigmatic. The exact roles of phosphoinositides in regulating autophagosome–lysosome fusion are less well understood, although PtdIns(3)P and PtdIns(4)P have been shown to be required. Similarly, less is known about how the movement of autophagosomes and lysosomes through microtubules and microfilaments is regulated in a timely manner and coordinates with autophagy activation. Most importantly, whether and how defects in these functional molecules interfere with autophagosome–lysosome fusion to participate in the development of human disease are largely unknown. In the future, further investigations are needed to determine the detailed functions of these factors involved in autophagosome–lysosome fusion and to explore the potential role(s) of deregulated autophagosome–lysosome fusion in human disease.

## Figures and Tables

**Figure 1 cells-13-00500-f001:**
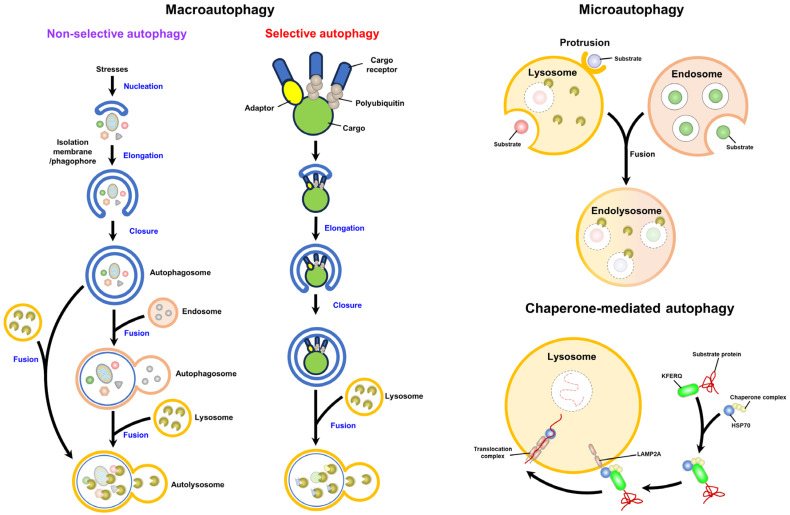
Types of autophagy. Three types of autophagy have been identified: macroautophagy, microautophagy, and chaperone-mediated autophagy. The process of macroautophagy involves stepwise vacuole biogenesis, including stress-induced nucleation of isolation membranes and subsequent elongation and closure into a double-membraned autophagosome. The mature autophagosome can fuse with the endosome to form an amphisome, which further fuses with the lysosome, becoming an autolysosome, in which lysosomal hydrolases degrade sequestered materials. The autolysosome can also be generated by direct fusion of the autophagosome with a lysosome. In addition to non-selective autophagy, macroautophagy may selectively target cargo through the specific cargo receptor-mediated recognition of cargo ubiquitination and cargo-binding adaptors. Microautophagy is a lysosomal process in which cytosolic materials are directly engulfed into the lysosomal lumen for degradation through lysosomal membrane invagination and transport via the ESCRT-mediated pathway. Chaperone-mediated autophagy is initiated by the HSC70-mediated recognition of degradative substrates containing KFERQ motifs and subsequent delivery to the lumen of lysosomes for degradation by docking with LAMP2A on lysosomal membranes.

**Figure 2 cells-13-00500-f002:**
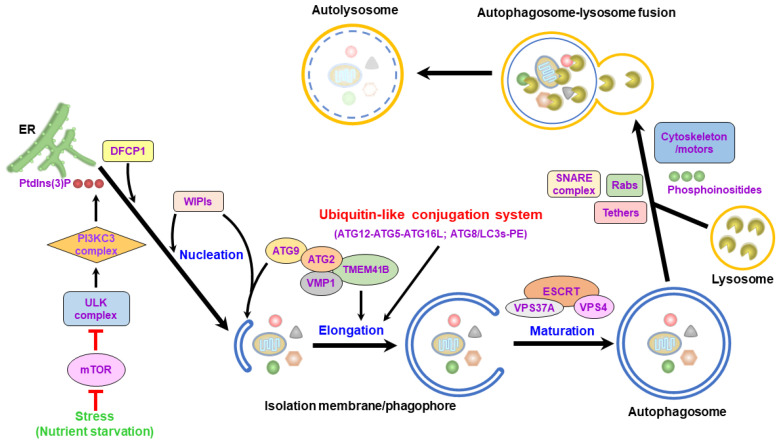
Autophagy process. Stresses, such as nutrient starvation, suppress mTOR activity, inducing the translocation of the ULK complex to the ER. The ER-localized ULK complex recruits and activates the PI3KC3 complex to generate PtdIns(3)P. PtdIns(3)P induces the recruitment of DFCP1 and WIPIs for the nucleation of isolation membranes/phagophores from the ER. The expansion of the isolation membrane/phagophore is elongated by lipid transfer mediated by ATG2, which can form a heteromeric complex with ATG9, VMP1, and TMEM41B. Two ubiquitin-like conjugates (ATG12-ATG12-ATG16L; ATG8/LC3s-PE) are also required for isolation membrane/phagophore elongation. The closure of the isolation membrane/phagophore to a double-membraned autophagosome is mediated by the ESCRT complex and its associated VPS4 and VPS37A. Mature autophagosomes then fuse with lysosomes to mature into autolysosomes. Several factors promote autophagosome–lysosome fusion, including the Rab family of small GTPases, SNARE complexes, tethering factors, cytoskeleton/associated motors, and phosphoinositides. Lysosomal hydrolases degrade the sequestered components within autolysosomes to recycle nutrients and regenerate energy.

**Figure 3 cells-13-00500-f003:**
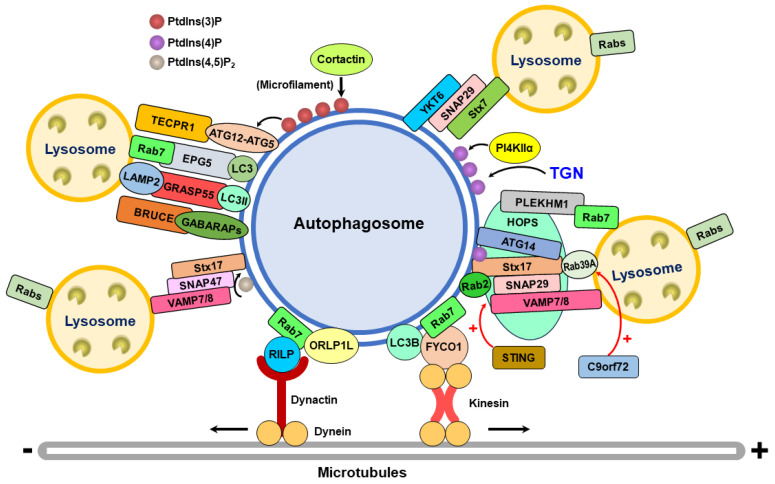
Summary of functional molecules involved in autophagosome–lysosome fusion. Several regulators function in the fusion of autophagosomes and lysosomes. Three types of SNARE complexes, Stx17-SNAP29-VAMP8, Stx17-SNAP29-VAMP8, and YKT6-SNAP29-Stx7, assemble at fusion sites to bridge the association of autophagosomes with lysosomes. Tethering factors, such as the HOPS complex and ATG14, are recruited by interactions with PLEKHM1 and the SNARE complex, thereby promoting lipid and content mixing and fusion. The other functional tethers for autophagosome–lysosome fusion, TECRP1, EPG5, GRASP55, and BRUCE, could be associated with autophagosomes and lysosomes through interactions with autophagosomal ATG8/LC3, the ATG12-ATG5 conjugate, lysosomal Rab7, and LAMP2. For the transport of autophagosomes for fusion to lysosomes, RILP and ORLP1L interact and bind to Rab7 on autophagosomes, and RILP interacts with the dynein–dynactin complex, facilitating minus-end-directed movement on microtubules. FYCO1 binds to Rab7 and LC3B and assembles into an adaptor complex, allowing microtubule plus-end-directed transport of autophagosomes, presumably through kinesin. PtdIns(3)P, located on autophagosomes, promotes the recruitment of TECRP1 and the assembly of microfilaments via cortactin for autophagosome–lysosome fusion. PI4KIIα and generated PtdIns(4)P are translocated from the TGN to autophagosomes when autophagy is activated, where they promote the fusion of autophagosomes to lysosomes.

**Table 1 cells-13-00500-t001:** The Rab family proteins and associated factors that function in autophagosome–lysosome fusion.

Name	Characteristics and Biological Functions	Reference(s)
Rab7	1. Rab7 colocalizes with autophagic vacuoles. 2. The GTPase activity of Rab7 is required for late-stage autophagic vacuole formation.	[116]
Rab7 and UVRAG	1. UVRAG increases Rab7 GTPase activity. 2. UVRAG enhances the colocalization of Rab7 with the autophagosome. 3. UVRAG promotes the recruitment of the Rab7 effector, RILP, to late endosomes and lysosomes for transport, thus facilitating autophagosome–lysosome fusion.	[67]
Rab7 and PLEKHM1	1. PLEKHM1 interacts with Rab7. 2. PLEKHM1 binds to the HOPS complex and ATG8/LC3 family proteins, facilitating autophagosome–lysosome fusion.	[123,158]
Rab7 and PLEKHM1	1. PtdIns(4,5)P2 promotes the conversion of GTP-bound Rab7 to GDP-bound Rab7. 2. PtdIns(4,5)P2 facilitates the release of Rab7 from PLEKHM1, thereby suppressing the fusion of autophagosomes with lysosomes.	[159]
Rab7 and Rubicon	1. Rubicon interacts with Rab7. 2. Rubicon forms a complex with the PI3KC3 complex, a process required for autophagy initiation. 3. Rubicon suppresses the fusion between autophagosomes and lysosomes.	[68,158]
Rab7 and NRBF2	1. NRBF2 interacts with the Rab7 GEF complex CCZ1-MONA1. 2. NRBF2 activates Rab7. 3. NRBF2 promotes autophagosome–lysosome fusion.	[160]
Rab2	1. The GTP-bound Rab2 colocalizes with Rab7 in starved *Drosophila melanogaster* cells. 2. Overexpression of GTP-bound Rab2 induces the accumulation of degradative autolysosomes in starved *Drosophila melanogaster* cells. 3. Gene silencing of Rab2A in MDA-MB-231 cells leads to the formation of LC3- and LAMP1-positive amphisomes.	[160,161]
Rab2	1. Torin1-mediated autophagy activation induces Rab2 translocation from the Golgi apparatus to autophagic vacuoles. 2. Rab2 can interact with ULK1 and ATG13. 3. Overexpression of Rab2 triggers autophagy initiation by promoting ULK1, ATG9, and ATG14 phosphorylations. 4. The binding of Rab2 to RUBUNL and Stx17 induces the formation of an autophagosomal trimeric complex, which then recruits the HOPS complex. 5. Gene silencing of Rab2 in Torin1-treated cells inhibits autolysosome formation and represses the colocalization between LC3 and LAMP1.	[162]
Rab2 and Rab39A	1. Rab39 can directly interact with Stx17 and the HOPS complex components, VPS39 and VPS41. 2. Induced autophagy by Torin1 promotes the localization of Rab39A in autophagosomes and autolysosomes. 3. Gene silencing of Rab39A suppresses autophagic flux and induces LC3^+^ puncta accumulation. 4. Rab39A overexpression promotes the binding of Stx17 to VPS33A and enhances the interaction between Rab2 and VPS39. 5. Overexpression of Rab39A inhibits the binding of Rab7 to VPS39 and VPS41. 6. Rab39A promotes the assembly of the HOPS complex to facilitate the SNARE-mediated membrane fusion between proteoliposomes. 7. C9orf72 GEF activates Rab39A and thus enables the formation of the HOPS complex to promote autophagosome–lysosome fusion.	[112,163]
Rab33B and OATL1	1. OATL1 is an autophagosome-resident GAP of Rab33B. 2. OATL1 interacts with ATG8/LC3 family proteins. 3. OATL1 suppresses the fusion of autophagosomes with lysosomes.	[164]
Rab21 and VAMP8	1. Starvation induces the activation of MTMR3, a GEF of Rab21. 2. MTMR3 activates Rab21 and promotes its binding to VAMP8. 3. Starvation-activated MTMR3 and Rab21 promote the endolysosomal trafficking of VAMP8, facilitating autophagosome–lysosome fusion.	[165]
Rab24 and Rab7	1. Rab24 colocalizes with Rab7 in late endosomes and lysosomes. 2. Rab24 interacts with Rab7 and RILP. 3. Rab24 promotes endolysosomal degradation.	[166]

**Table 2 cells-13-00500-t002:** SNAREs and related regulators that function in autophagosome–lysosome fusion.

Name	Characteristics and Biological Functions	Reference(s)
VAMP7	1. VAMP7 localizes on autophagosomes. 2. The deletion of the R-SNARE motif and TMD of VAMP7 inhibits autolysosome maturation. 3. VAMP7 is required for autophagosome–lysosome fusion.	[101]
Vit1b and VAMP8	1. Vit1b and VAMP8 colocalize on autophagosomes. 2. Gene silencing of Vit1b and VAMP8 represses the fusion between autophagosomes and lysosomes and increases the number of autophagosomes. 3. Vit1b and VAMP8 positively regulate the fusion of autophagosomes with lysosomes.	[184]
Stx17-SNAP29-VAMP8 complex	1. Gene knockdown of Stx17 induces autophagosome accumulation. 2. Stx17 interacts with VAMP8. 3. SNAP29 enhances the interaction of Stx17 with VAMP8. 4. Gene silencing of Stx17 and SNAP29 induces the accumulation of autophagosomes and suppresses autophagic flux. 5. Stx17, SNAP29, and VAMP8 form a SNARE complex for autophagosome–lysosome fusion.	[185]
SCFD1 and VAMP8	1. SCFD1 binds to Stx17 and VAMP8 of the SNARE complex. 2. mTORC1-phosphorylated VAMP8 inhibits the interaction of SCFD1 with the SNARE complex. 3. Gene knockdown of SCFD1 interferes with the fusion of autophagosomes with lysosomes. 4. SCFD1 forms a complex with the Stx17-SNAP29-VAMP8 SNARE complex, promoting the fusion between autophagosomes and lysosomes.	[186]
SCFD1 and VAMP8	1. SCFD1 can be acetylated and deacetylated by KAT2B/PCAF and SIRT4, respectively. 2. Starvation activates AMPK to phosphorylate SCFD1, leading to the deacetylation of SCFD1. 3. SCFD1 deacetylation promotes its binding to the Stx17-SNAP29-VAMP8 SNARE complex. 4. Acetylation and deacetylation regulate the function of SCFD1 in regulating autophagosome–lysosome fusion.	[187]
Stx17 and the HOPS complex	1. Stx17 interacts with the HOPS complex. 2. Gene knockdown of the components of the HOPS complex suppresses autophagic flux. 3. The HOPS complex promotes autophagosome–lysosome fusion by interacting with Stx17.	[122]
Stx17-SNAP29-VAMP7 complex	1. Stx17, SNAP29, and VAMP7, rather than VAMP8, form a heteromeric complex. 2. The phosphorylation of Stx17 at Ser2 suppresses the interaction between Stx17 and VAMP7. 3. The Ser2 phosphorylation of Stx17 differentially regulates the binding of Stx17 to VPS33A, thus switching prefusion to fusion at the autophagosome–lysosome fusion stage.	[188]
Stx17-SNAP29-VAMP8 complex	1. Stx17 is acetylated at Lys219 and Lys223 by CBP and deacetylated by HDAC2. 2. Activation of autophagy leads to Stx17 deacetylation. 3. The deacetylation of Stx17 promotes its interaction with SNAP for the assembly of the Stx17-SNAP29-VAMP8 complex and enhances its binding to the HOPS complex. 3. Stx17 deacetylation positively regulates autophagosome–lysosome fusion.	[189]
STING and Stx17-SNAP29-VAMP8 complex	1. STING directly binds to Stx17 through its C-terminal domain. 2. The binding of STING to Stx17 disrupts the assembly of the Stx17-SNAP29-VAMP8 complex. 2. The activation of STING by phosphorylation and polymerization relieves its inhibitory effect on the formation of the Stx17-SNAP29-VAMP8 complex. 3. The defect in STING degradation and the lupus-associated STING mutants interfere with autophagosome–lysosome fusion and repress autophagic flux.	[190]
ATG14 and Stx17	1. ATG14 interacts with Stx17 through the CCD of ATG14 and the SNARE motif of Stx17. 2. ATG14 harbors a tethering ability to promote lipid mixing in the protein-free liposome membranes. 3. ATG14 promotes lipid mixing and content mixing of proteoliposomes containing Stx17, SNAP29, and VAMP8. 4. Interference with the Stx17 and ATG14 interactions represses membrane fusion. 5. The Stx17-SNAP29-VAMP8 SNARE complex promotes autophagosome–lysosome fusion through ATG14.	[121]
SNAP29 O-GlcNAcylation	1. OGT induces the O-GlcNAcylation of SNAP29 at Ser2, Ser61, Thr130, and Ser153. 2. SNAP29 O-GlcNAcylation inhibits the interaction between Stx17 and VAMP8. 3. Gene silencing of OGT promotes the assembly of the Stx17-SNAP29-VAMP8 SNARE complex. 4. Glucose and nutrient deprivation reduce SNAP29 O-GlcNAcylation and promote the formation of the Stx17-SNAP29-VAMP8 SNARE complex for autophagosome–lysosome fusion.	[191]
SNAP29 O-GlcNAcylation	1. SM15 induces the O-GlcNAcylation of SNAP29 to inhibit SNARE complex assembly and suppress autophagic flux. 2. SM15-mediated SNAP29 O-GlcNAcylation promotes ROS-induced cell apoptosis.	[192]
YKT6	1. YKT6 localizes on autophagosomes. 2. Gene silencing of YKT6 inhibits the maturation of autolysosomes. 3. YKT6 interacts with Stx7 and SNAP29, forming a SNARE complex. 4. YKT6 can promote in vitro autophagosome–vacuole fusion. 5. YKT6 is another v-SNARE in addition to Stx17 for autophagosome–lysosome fusion.	[193,194]
YKT6 phosphorylation	1. ULK1 phosphorylates YKT6 at Thr156. 2. Increased YKT Thr156 phosphorylation interferes with the binding of YKT6 to SNAP29 to block the fusion of autophagosomes to lysosomes. 3. The ULK1-induced phosphorylation of YTK6 on autophagosomes prevents the immature formation of the SNARE complex for autophagosome–lysosome fusion.	[195]
Stx17-SNAP47-VAMP7/VAMP8 complex	1. Mitophagy induction promotes the localization of SNAP47 in mitophagosomes. 2. Gene knockout of SNAP47 interferes with O/A- and hypoxia-induced mitophagic degradation. 3. SNAP47 interacts with Stx17 and VAMP7/VAMP8 to form the functional SNARE complex for autophagosome–lysosome fusion in mitophagy. 4. SNAP47 is recruited to autophagosomes by PtdIns(4, 5)P2 and ATG8/LC3s. 5. SNAP47 could play a functional role with SNAP29 in the fusion of autophagosomes with lysosomes in starvation-induced autophagy.	[196]

**Table 3 cells-13-00500-t003:** The tethering factors required for autophagosome–lysosome fusion.

Name	Characteristics and Biological Functions	Reference(s)
The HOPS complex	1. The HOPS complex interacts with Stx17. 2. Gene silencing of the components of the HOPS complex represses autophagic flux. 3. The HOPS complex facilitates autophagosome–lysosome fusion by binding to Stx17. 4. VPS33A can differentially bind to Stx17 in a Stx17 Ser2 phosphorylation-dependent manner, regulating the switch from the prefusion state to the fusion state of autophagosome–lysosome fusion.	[122,188]
The HOPS complex and PLEKHM1	1. PLEKHM1 interacts with VPS39 and VPS41 of the HOPS complex. 2. The HOPS complex is recruited to the membrane fusion site by interacting with PLEKHM1, thus promoting autophagosome–lysosome fusion.	[123]
The HOPS complex and Pacer	1. Pacer stimulates PI3KC3 complex activity by antagonizing Rubicon. 2. Pacer promotes the recruitment of the PI3KC3 and HOPS complexes on autophagosomes via Stx17 and phosphoinositides. 3. The HOPS complex facilitates the fusion between autophagosomes and lysosomes through Stx17 and Pacer.	[204]
The HOPS complex and Rab39A	1. Rab39A triggers the formation of the HOPS complex to promote SNARE-mediated membrane fusion. 2. C9orf72 GEF activates Rab39A to drive the assembly of the HOPS complex, thus facilitating autophagosome–lysosome fusion.	[112]
ATG14	1. The CCD of ATG14 binds to the SNARE motif of Stx17. 2. ATG14 alone could be a membrane tether that induces lipid mixing of protein-free liposome membranes. 3. ATG14 triggers lipid mixing and content mixing of proteoliposomes reconstituted with Stx17, SNAP29, and VAMP8. 4. Disruption of the binding of ATG14 to Stx17 suppresses membrane fusion. 5. ATG14 facilitates autophagosome fusion with the lysosome by interacting with the Stx17-SNAP29-VAMP8 SNARE complex.	[121]
EPG5	1. EPG5 interacts with Rab7 on late endosomes, lysosomes, and autophagosomes. 2. EPG5 promotes the assembly of the Stx17-SNAP29-VAMP8 SNARE complex. 3. EPG5 is a membrane tether that induces the lipid mixing of proteoliposomes through Stx17-SNAP29-VAMP7. 4. EPG5 promotes autophagosome–lysosome fusion.	[126]
GRASP55	1. GRASP55 interacts with LC3-II on autophagosomes and LAMP2 on lysosomes. 2. Glucose deprivation induces GRASP55 de-O-GlcNAcylation, promoting the translocation of GRASP55 to autophagosomes. 3. GRASP55 could be a membrane tethering factor for the fusion of autophagosomes with lysosomes.	[205]
TECPR1	1. TECPR1 interacts with the ATG12-ATG5 conjugate, recruiting the ATG12-ATG5 conjugate to autolysosomes. 2. TECRP1 binds to PtdIns(3)P in an ATG12-ATG5 complex-dependent manner. 3. TECPR1 could serve as a membrane tether for autophagosome–lysosome fusion.	[206]
BRUCE	1. BRUCE interacts with the Stx17, GABARAP, and GABARAPL proteins. 2. BRUCE promotes autophagosome–lysosome fusion via its tethering activity.	[207]

**Table 4 cells-13-00500-t004:** Phosphatidylinositol phosphates (phosphoinositides) that function in autophagosome–lysosome fusion.

Name	Characteristics and Biological Functions	Reference(s)
PtdIns(3)P	1. TECRP1 binds to PtdIns(3)P and associates with the ATG12-ATG5 conjugate. 2. PtdIns(3)P is required for the membrane tethering function of TECRP1.	[206]
PtdIns(3,5)P2	1. PIKFYVE phosphorylates PtdIns(3)P to produce PtdIns(3,5)P2. 2. PtdIns(3,5)P2 can be converted to PtdIns(3)P by INPP5E. 3. Gene knockdown of INPP5E induces autophagosome accumulation and inhibits autophagic flux. 4. The INPP5E-catalyzed conversion of PtdIns(3,5)P2 to PtdIns(3)P promotes microfilament polymerization by activating cortactin.	[212,213]
PtdIns(4)P	1. PI4KIIα converts PtdIns to PtdIns(4)P. 2. Nutrient starvation induces the translocation of PI4KIIα and PtdIns(4)P from the TGN to autophagosomes. 3. Gene knockdown of PI4KIIα suppresses the fusion of autophagosomes with lysosomes and induces autophagosome accumulation. 4. GABARAPs but not LC3s promote PI4KIIα recruitment to autophagosomes. 5. PI4KIIα and PtdIns(4)P facilitate autophagosome–lysosome fusion.	[214]
PtdIns(4)P	1. Stx17 interacts with PtdIns(4)P through its C-terminal positive-charged amino acids and colocalizes with PtdIns(4)P on autophagosomes. 2. Gene knockdown of PI4KIIα and treatment of the PI4KIIα inhibitor impair the recruitment of Stx17 to autophagosomes and block autophagosome–lysosome fusion.	[215]
PtdIns(4, 5)P2	1. PIP5Kγ promotes the conversion of PtdIns(4)P to PtdIns(4, 5)P2 on late endosomes. 2. PtdIns(4, 5)P2 inactivates Rab7 and dissociates PLEKHM1 from late endosomes and lysosomes. 3. PIP5Kγ-catalyzed PtdIns(4, 5)P2 may drive the recycling of Rab 7 and PLEKHM1 on late endosomes and lysosomes for autophagosome–lysosome fusion.	[159]
PtdIns(4, 5)P2	1. PtdIns(4, 5)P2 promotes the recruitment of SNAP47 on autophagosomes. 2. The PtdIns(4, 5)P2-mediated autophagosomal translocated SNAP47 interacts with Stx17 and VAMP7/VAMP8, forming a ternary SNARE complex to regulate autophagosome–lysosome fusion in mitophagy.	[196]

**Table 5 cells-13-00500-t005:** Cytoskeleton and associated motor proteins that function in autophagosome–lysosome fusion.

Name	Characteristics and Biological Functions	Reference(s)
Microtubules and dynein/dynactin	1. Autophagosomes move toward the γ-tubulin-positive centrosome. 2. The microinjection of anti-LC3 antibodies interferes with the fusion of autophagosomes with lysosomes. 3. The microinjection of an anti-dynactin antibody and overexpression of dynamitin block the perinuclear localization of autophagosomes. 4. Dynein–dynactin and minus-end-directed movement on microtubules are required to transport autophagosomes for fusion with lysosomes.	[221]
RILP and dynein/dynactin	1. RILP interacts with the dynein–dynactin complex. 2. RILP bridges the association between the dynein–dynactin complex and autophagosomes via LC3 and the LIR of RILP. 3. Gene knockdown of RILP inhibits the movement of autophagosomes and lysosomes. 4. RILP is a mediator for transporting autophagosomes and lysosomes through interactions with LC3 and Rab7.	[119,168,222]
ORP1L and RILP	1. The intracellular amount of cholesterol regulates the perinuclear localization of autophagosomes. 2. The cholesterol sensor ORP1L can bind to RILP. 3. Gene silencing of ORP1L restores the perinuclear localization of autophagosomes in cholesterol-depleted cells. 4. ORP1L regulates the minus-end-directed movement of autophagosomes and the recruitment of PLEKHM1 and HOPS complexes to late endosomes and lysosomes. 5. ORP1L and RILP binding promote autophagosome–lysosome fusion.	[130,224]
FYCO1	1. FYCO1 is an effector of Rab7 that can bind to LC3B. 2. FYCO1 interacts with LC3B via its PtdIns(3)P-binding FYVE motif. 3. FYCO1 is recruited to late endosomes and lysosomes through binding to Rab7. 4. FYCO1 forms an adaptor complex with Rab7 and LC3B and competes with RILP for binding to Rab7, facilitating the plus-end-directed movement of autophagosomes on microtubules. 5. The STK4-mediated phosphorylation of LC3B at Thr50 by nutrient starvation suppresses the interaction of LC3B with FYCO1, facilitating the perinuclear localization of autophagosomes.	[118]
Microfilaments	1. HDAC6 promotes autophagosome–lysosome fusion via F-actin polymerization. 2. Gene knockout of HDAC6 suppresses autophagosome fusion with the lysosome and induces autophagosome accumulation. 3. HDAC6 recruits cortactin to assemble microfilaments to facilitate the fusion between autophagosomes and lysosomes.	[131]
Microfilaments and myosin VI	1. Myosin VI can be recruited to autophagosomes and interacts with NDP52, Optineurin, and T6BP. 2. Myosin VI binds to Tom1/Tom1L2, mediating the delivery of endosomal contents to autophagosomes. 3. Gene knockdown of Tom1/Tom1L2 suppresses the fusion between autophagosomes and lysosomes. 4. Myosin-Tom1/Tom1L2 positively regulates autophagosome–lysosome fusion.	[225]
Microfilaments and KRT8	1. KRT8 binds to microfilaments through PLEC. 2. KRT8 stabilizes actin filaments, thereby facilitating autophagosome–lysosome fusion.	[226]

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
