# Peer review of "Molecular Mechanism of Autophagosome–Lysosome Fusion in Mammalian Cells"

_cells, 2024, doi:10.3390/cells13060500_

Round 1
Reviewer 1 Report
Comments and Suggestions for Authors
The manuscript aims to summarize the molecular mechanisms of autophagosome-lysosome fusion. The approach taken is a list-like description of results from individual papers dealing with this topic. This approach is very detailed and comprehensive, but it also makes reading difficult and confusing. The Tables are also listing the results paper-by-paper, rather than protein-by-protein.
1. Clearly, a more analytical approach would be more reader-friendly. A good example of an analytical approach to this same topic is a very recent review by Jiajie Diao, Calvin K. Yip & Qing Zhong, published in the Open Access journal Autophagy Reports (DOI: 10.1080/27694127.2024.2305594).
2. In addition to autophagosome-lysosome fusion, autophagosome-endosome fusion should also be described and referenced. The molecular requirements of these two fusion steps differ.
3. Phagophore fusion to form a closed autophagosome needs additional factors in addition to the ATG-proteins that also assist phagophore elongation. The proteins known to function in phagophore closure should also be mentioned in text the Figure 1.
4. The author should check that all recent and relevant papers are cited, e.g. Autophagy. 2023;19:2078–2093; J Cell Sci. 2023;136: jcs260546; Cell Rep. 2023;42:111969
5. A schematic figure on the different types of autophagy would be helpful.
6. It is a bit confusing that the Introduction starts by explaining microautophagy and CMA, taking into account that the topic of the review is autophagosome-lysosome fusion. It would be more logical to start by explaining macroautophagy first.
Minor comments:
Line 220: “member-anchoring” – should it be membrane-anchoring?
Line 242: “nutrient-starved HeLa cells starved” – remove the latter ‘starved’
Line 277: “HOPS–tethering 277 complex, which contains vacuolar protein sorting 33A (VPS33A), VPS16, and VPS39” this sentence is a bit misleading. Please list all subunits of the HOPS complex.
Table 4: “GABARPs but not LC3s ” -> GABARAPs but not LC3s
Lines 477 and 480: RIPL should be RILP
Line 543: “and LAMP2 on autophagosomes and lysosomes”. LAMP2 is not present on autophagosomes, please reword this.
Line 556: “autophagosome-lysosome function” - should this be autophagosome-lysosome fusion?
Lines 579-580: “Additionally, at least three SNARE complexes function in the fusion of autophagosomes to lysosomes have been identified” - > Additionally, at least three SNARE complexes that function in the fusion of autophagosomes to lysosomes have been identified
Comments on the Quality of English Languagesee comments to the author
Author Response
Dear reviewer:
Thank you for giving me the opportunity to resubmit my manuscript “Molecular Mechanism of Autophagosome-Lysosome Fusion in Mammalian Cells” to Cells (Manuscript ID: cells-2889607). I appreciate the thoughtful and constructive suggestions provided by the reviewers. The content of this manuscript has been improved based on the reviewers’ comments. The changes are shown in the revised manuscript, and point-by-point responses to each comment are listed below.
Point 1: The manuscript aims to summarize the molecular mechanisms of autophagosome-lysosome fusion. The approach taken is a list-like description of results from individual papers dealing with this topic. This approach is very detailed and comprehensive, but it also makes reading difficult and confusing. The Tables are also listing the results paper-by-paper, rather than protein-by-protein. Clearly, a more analytical approach would be more reader-friendly. A good example of an analytical approach to this same topic is a very recent review by Jiajie Diao, Calvin K. Yip & Qing Zhong, published in the Open Access journal Autophagy Reports (DOI: 10.1080/27694127.2024.2305594).
Response 1: I thank the reviewer for the thoughtful comments on our manuscript and for recognizing the merits of our manuscript. Most previous review articles on autophagy have focused on briefly introducing the conclusions of different studies, and limited information on the detailed methods and few comparisons between different papers has been provided. I expect that this review article provides comprehensive information on the regulation of autophagosome-lysosome fusion based on the conclusions of different studies. Additionally, I intend to provide compelling information about the history of discovery and the molecular action of functional molecules involved in autophagosome-lysosome fusion. The information in these tables will also allow the readers to easily determine the biological functions of these molecules in regulating the fusion between autophagosomes and lysosomes and their original studies. I have incorporated the analytic part for these functional molecules in the last paragraph of each part of section 3 (sections 3.1~3.5). In these paragraphs of section 3.1~section 3.5, I summarize the main concepts related to the biological significance of these functional molecules in the regulation of autophagosome-lysosome fusion and analytically propose potential questions that remain to be answered. Please see lines 252~264 of paragraph 3 on page 7, lines 417~430 of paragraph 2 on page 12, lines 483~489 of paragraph 3 on page 14, lines 535~543 of paragraph 2 on page 16, and line 633 of paragraph 4 on page 18 to line 641 of paragraph 1 on page 19 in the revised manuscript. It is hoped that the reviewer will kindly agree with me to maintain the completeness of the formatting and original idea for this review article in the revised manuscript. Thank you again for the thoughtful suggestions.
Point 2: In addition to autophagosome-lysosome fusion, autophagosome-endosome fusion should also be described and referenced. The molecular requirements of these two fusion steps differ.
Response 2: Thank you for your thoughtful suggestions. In the revised manuscript, I have added a new paragraph to introduce autophagosome-endosome fusion in section 2. Please see lines 133-147 of paragraph 2 on page 4 in the revised manuscript.
Point 3: Phagophore fusion to form a closed autophagosome needs additional factors in addition to the ATG-proteins that also assist phagophore elongation. The proteins known to function in phagophore closure should also be mentioned in text the Figure 1.
Response 3: I am very grateful for the reviewer’s thoughtful comment. In the revised manuscript, I have incorporated the information on the functional factors required for the elongation of the isolation membrane/phagophore and the closure into autophagosomes in section 2 and Figure 2. Please see line 116 of paragraph 2 on page 3 to line 132 of paragraph 1 on page 4 and Figure 2 on page 3 in the revised manuscript.
Point 4: The author should check that all recent and relevant papers are cited, e.g. Autophagy. 2023;19:2078–2093; J Cell Sci. 2023;136: jcs260546; Cell Rep. 2023;42:111969.
Response 4: I thank the reviewer for this comment. I have incorporated the information from these studies in section 3.2 and section 3.5 in the revised manuscript. Please see lines 395~399 of paragraph 3 on page 11, lines 410~416 of paragraph 1 on page 12, lines 567~572 of paragraph 3 on page 16, and Table 2 on pages 9~10 in the revised manuscript.
Point 5: A schematic figure on the different types of autophagy would be helpful.
Response 5: I thank the reviewer for the thoughtful comments. A schematic diagram showing three types of autophagy, macroautophagy (non-selective and selective autophagy), microautophagy, and chaperone-mediated autophagy, has been included in the revised manuscript. Please see Figure 1 on page 2 in the revised manuscript.
Point 6: It is a bit confusing that the Introduction starts by explaining microautophagy and CMA, taking into account that the topic of the review is autophagosome-lysosome fusion. It would be more logical to start by explaining macroautophagy first.
Response 6: Thank you for your thoughtful suggestions. I have revised the content of section 1 in the revised manuscript to introduce macroautophagy first and then to introduce microautophagy and chaperone-mediated autophagy. Please see lines 28~41 of paragraph 2 on page 1 in the revised manuscript.
Point 7: Line 220: “member-anchoring” – should it be membrane-anchoring?
Response 7: I appreciate the reviewer for this correction. The term “member-anchoring” has been corrected to “membrane-anchoring”. Please see lines 283~284 of paragraph 4 on page 7 in the revised manuscript.
Point 8: Line 242: “nutrient-starved HeLa cells starved” – remove the latter ‘starved’.
Response 8: I thank the reviewer for this correction. The “nutrient-starved HeLa cells starved” has been corrected to “nutrient-starved HeLa cells”. Please see line 306 of paragraph 2 on page 8 in the revised manuscript.
Point 9: Line 277: “HOPS–tethering 277 complex, which contains vacuolar protein sorting 33A (VPS33A), VPS16, and VPS39” this sentence is a bit misleading. Please list all subunits of the HOPS complex.
Response 9: I appreciate the reviewer for this comment. I have revised this sentence to describe all the components of the HOPS complex in the revised manuscript. Please see lines 436~437 of paragraph 3 on page 12 in the revised manuscript.
Point 10: Table 4: “GABARPs but not LC3s ” -> GABARAPs but not LC3s.
Response 10: I thank the reviewer for this correction. The term “GABARPs but not LC3s” has been corrected to “GABARAPs but not LC3s” in Table 4 in the revised manuscript. Please see Table 4 on page 15 in the revised manuscript.
Point 11: Lines 477 and 480: RIPL should be RILP.
Response 11: I appreciate the reviewer for this correction. The term “RIPL” has been corrected to “RILP”. Please see lines 585 and 588 of paragraph 1 on page 18 in the revised manuscript.
Point 12: Line 543: “and LAMP2 on autophagosomes and lysosomes”. LAMP2 is not present on autophagosomes, please reword this.
Response 12: I thank the reviewer for this correction. This sentence has been revised to “…ATG12-ATG5 conjugate, and ATG8/LC3s on autophagosomes, and Rab7 and LAMP2 on lysosomes”. Please see lines 658 and 659 of paragraph 2 on page 19 in the revised manuscript.
Point 13: Line 556: “autophagosome-lysosome function” - should this be autophagosome-lysosome fusion?
Response 13: I appreciate the reviewer for this correction. The term “autophagosome-lysosome function” has been corrected to “autophagosome-lysosome fusion” in the revised manuscript. Please see line 672 of paragraph 2 on page 19 in the revised manuscript.
Point 14: Lines 579-580: “Additionally, at least three SNARE complexes function in the fusion of autophagosomes to lysosomes have been identified” - > Additionally, at least three SNARE complexes that function in the fusion of autophagosomes to lysosomes have been identified.
Response 14: I thank the reviewer for this correction. This sentence has been corrected to “Additionally, at least three SNARE complexes that function in the fusion of autophagosomes to lysosomes have been identified”, as the reviewer suggested. Please see lines 734~735 of paragraph 1 on page 21 in the revised manuscript.
We hope that this version of our manuscript and our responses address all your concerns and that this revised manuscript meets the criteria for publication in Cells. Thank you for your kind consideration.
Sincerely,
Po-Yuan Ke, Ph.D.
Associate Professor
Department of Biochemistry & Molecular Biology and Graduate Institute of Biomedical Sciences, College of Medicine, Chang Gung University, Taoyuan 33302, Taiwan, Republic of China
Liver Research Center, Chang Gung Memorial Hospital, Linkou, Taoyuan 33305, Taiwan, Republic of China
Tel: 886-3-2118800-5115
E-mail: pyke0324@mail.cgu.edu.tw

Reviewer 2 Report
Comments and Suggestions for Authors
This review is well written and contains valuable information for the autophagy researchers. However, I recommend the author to revise the manuscript for the publication.
1. Lines 40-42; microautophagy found in mammalian cells, called endosomal microautophagy requires fusion of late endosomes with lysosomes (reference 11). Therefore, it is wrong that “only macroautophagy requires fusion with lysosomes”.
2. Chapter 3.6 Summary is just the repetitive description of 3.1~3.5. Because the figure 2 summarizes these contents, this paragraph is not necessary. The author needs to refer to figure 2 at every chapters.
3. There are numerous reports indicating the autophagosome accumulation in various disease model cells. I consider that the disease-associated changes in factors involved in autophagosome-lysosome fusion would be reported. If possible, I recommend to add the information concerning the relationship between this fusion and the disease-related accumulation of autophagosomes.
Author Response
Dear reviewer:
Thank you for giving me the opportunity to resubmit my manuscript “Molecular Mechanism of Autophagosome-Lysosome Fusion in Mammalian Cells” to Cells (Manuscript ID: cells-2889607). I appreciate the thoughtful and constructive suggestions provided by the reviewers. The content of this manuscript has been improved based on the reviewers’ comments. The changes are shown in the revised manuscript, and point-by-point responses to each comment are listed below.
Point 1: Lines 40-42; microautophagy found in mammalian cells, called endosomal microautophagy requires fusion of late endosomes with lysosomes (reference 11). Therefore, it is wrong that “only macroautophagy requires fusion with lysosomes”.
Response 1: Thank you very much for the thoughtful suggestions. As suggested by the reviewer, I have deleted the sentence “only macroautophagy requires fusion with lysosomes” in the revised manuscript. Please see lines 41~42 of paragraph 2 on page 1 in the revised manuscript.
Point 2: Chapter 3.6 Summary is just the repetitive description of 3.1~3.5. Because the figure 2 summarizes these contents, this paragraph is not necessary. The author needs to refer to figure 2 at every chapters.
Response 2: I appreciate the reviewer for the constructive suggestions. In section 3.6, I intend to briefly outline the regulation of autophagosome-lysosome fusion by these identified functional molecules and summarize their actions in a schematic diagram (Figure 3 in the revised manuscript). I have incorporated the analytic part for these functional molecules in the last paragraph of each part of section 3 (sections 3.1~3.5). In these paragraphs of section 3.1~section 3.5, I summarize the biological significance of these functional molecules in regulating autophagosome-lysosome fusion and refer to Figure 3. In addition, I analytically propose potential questions that remain to be answered regarding the regulation of autophagosome-lysosome fusion in the future in the revised manuscript. Please see lines 252~264 of paragraph 3 on page 7, lines 417~430 of paragraph 2 on page 12, lines 483~489 of paragraph 3 on page 14, lines 535~543 of paragraph 2 on page 16, and line 633 of paragraph 4 on page 18 to line 641 of paragraph 1 on page 19 in the revised manuscript. It is hoped that the reviewer will kindly agree with me to keep section 3.6 for the competence of this review article in the revised manuscript. Thank you again for the thoughtful comments.
Point 3: There are numerous reports indicating the autophagosome accumulation in various disease model cells. I consider that the disease-associated changes in factors involved in autophagosome-lysosome fusion would be reported. If possible, I recommend to add the information concerning the relationship between this fusion and the disease-related accumulation of autophagosomes.
Response 3: Thank you very much for the thoughtful comments. I have incorporated section 3.7 in the revised manuscript to outline the studies showing how the deregulation of autophagosome-lysosome fusion is associated with the development of human diseases, including neurodegenerative diseases, myopathies, sepsis, encephalopathy, and viral infection. Please see line 674 of paragraph 3 on page 19 to line 712 of paragraph 2 on page 20 in the revised manuscript. Thank you again for this suggestion.
We hope that this version of our manuscript and our responses address all your concerns and that this revised manuscript meets the criteria for publication in Cells. Thank you for your kind consideration.
Sincerely,
Po-Yuan Ke, Ph.D.
Associate Professor
Department of Biochemistry & Molecular Biology and Graduate Institute of Biomedical Sciences, College of Medicine, Chang Gung University, Taoyuan 33302, Taiwan, Republic of China
Liver Research Center, Chang Gung Memorial Hospital, Linkou, Taoyuan 33305, Taiwan, Republic of China
Tel: 886-3-2118800-5115
E-mail: pyke0324@mail.cgu.edu.tw

Round 2
Reviewer 1 Report
Comments and Suggestions for Authors
The revision has addressed my previous comments only partially. I also mention below some new comments, mainly concerning the updated parts of the manuscript.
1. In my previous comments, I requested that the author should check that all recent and relevant papers are cited. However, relevant results and references are still missing:
- Role of Rab2 in autophagosome-lysosome fusion (e.g. PMID: 28483915, PMID: 30957628, PMID: 38083843, PMID: 37443788)
-Role of PI4P in autophagosome-lysosome fusion: recruitment of STX17 (PMID: 38411137) – this is a very recent paper but would fit perfectly to the review
-Role of SNAP47 in autophagosome-lysosome fusion (PMID: 38182888) – this is also a very recent paper but would fit perfectly to the review
2. I also commented on missing information of autophagosome-endosome fusion. This request was not given enough attention. Even though less details are known on autophagosome-endosome fusion (compared to autophagosome-lysosome fusion), some results have been published. Here are two examples of articles on the molecular mechanism of autophagosome-endosome fusion (note that this list is not complete):
- Autophagy 2019 Jan;15(1):34-5: ZFYVE26 mutations lead to defects in the fusion between autophagosomes and endosomes
- Molecular Biology of the Cell 2024, 35:ar40, 1–24: amphisomes can form in absence of HOPS complex components
3. Abstract, lines 33-34 and 41 in Introduction, Legend to Figure 1, and line 156: ”delivery to autolysosomes for degradation by acidic proteases” Degradation of autophagic cargo requires many types of lysosomal enzymes in addition to proteases (to degrade lipids, carbohydrates, nucleic acids, to remove posttranslational modifications, etc). Please reword all the sentences referring to proteases accordingly.
4. In Figure 1, the schematic presentation of microautophagy is problematic. The present figure gives the impression that the large lysosome is internalizing the small endosome as a whole, while the correct presentation would be to indicate that the endosome fuses with the lysosome.
5. Line 149-150: “To date, less is known about how autophagosomes fuse with lysosomes to form amphisomes.” Fusion of autophagosome with lysosome makes autolysosome, not amphisome. Should there be ‘how autophagosomes fuse with endosomes’?
Minor
Line 272: ‘Raf family of’ should be ‘Rab family of’
Line 422: ‘ULK1-mediated YKT6 phosphorylation by the phospho-mimetic mutant (T156E) of YKT6’ should read ‘ULK1-mediated YKT6 phosphorylation and the phospho-mimetic mutant (T156E) of YKT6’
Line 431: SANP29 should be SNAP29
Line 580: movment should be movement
Line 550: Oppostively – I do not know this word. Should it be ‘oppositely’?
Author Response
Dear reviewer:
Thank you for giving me the opportunity to resubmit my manuscript “Molecular Mechanism of Autophagosome-Lysosome Fusion in Mammalian Cells” to Cells (Manuscript ID: cells-2889607). I am very grateful for your thoughtful and constructive suggestions. The content of this manuscript has been improved based on your comments. The changes are shown in the revised manuscript, and point-by-point responses to each comment are listed below.
Point 1: In my previous comments, I requested that the author should check that all recent and relevant papers are cited. However, relevant results and references are still missing:
- Role of Rab2 in autophagosome-lysosome fusion (e.g. PMID: 28483915, PMID: 30957628, PMID: 38083843, PMID: 37443788)
-Role of PI4P in autophagosome-lysosome fusion: recruitment of STX17 (PMID: 38411137) – this is a very recent paper but would fit perfectly to the review
-Role of SNAP47 in autophagosome-lysosome fusion (PMID: 38182888) – this is also a very recent paper but would fit perfectly to the review
Response 1: I thank the reviewer for the thoughtful comments on our manuscript. I have incorporated the information from these studies and updated the new findings from recent publications in section 3.1, section 3.2, and section 3.4 in the revised manuscript. Please see the changes as follows,
- Section 3.1: line 265 of paragraph 1 on page 8 to line 339 of paragraph 2 on page 9, Table 1 on pages 6~7, and Figure 3 on page 24 in the revised manuscript. (PMID: 32543313, PMID:37821429, PMID:28483915, PMID:30957628, PMID:38083843, PMID:37443788, PMID:25453831, PMID:27255086, PMID:28063257, PMID:18448665, PMID:32960676, PMID:29229996, PMID:26325487, PMID:38323995)
- Section 3.2: lines 454~463 of paragraph 2 on page 13, line 513 of paragraph 4 on page 14 to line 528 of paragraph 1 on page 15, Table 2 on pages 11~12, and Figure 3 on page 24 in the revised manuscript. (PMID: 23217709, PMID:34785650, PMID:35465820, PMID:30655294, PMID:32264736, PMID:25419848, PMID:36704963, PMID:30097515, PMID:29789439, PMID:36644903, PMID:38182888, PMID:24554770)
- Section 3.4: lines 640~647 of paragraph 1 on page 19, lines 653~656 of paragraph 2 on page 19, Table 4 on page 18, and Figure 3 on page 24 in the revised manuscript. (PMID: 27340123, PMID:26038556, PMID:38411137, PMID:38182888)
Point 2: I also commented on missing information of autophagosome-endosome fusion. This request was not given enough attention. Even though less details are known on autophagosome-endosome fusion (compared to autophagosome-lysosome fusion), some results have been published. Here are two examples of articles on the molecular mechanism of autophagosome-endosome fusion (note that this list is not complete):
- Autophagy 2019 Jan;15(1):34-5: ZFYVE26 mutations lead to defects in the fusion between autophagosomes and endosomes
- Molecular Biology of the Cell 2024, 35:ar40, 1–24: amphisomes can form in absence of HOPS complex components
Response 2: I am very grateful for the reviewer’s thoughtful comment. In the revised manuscript, I have incorporated the compelling information on in section 2. Related studies, including these two papers as the reviewer suggested, are included. Please see lines 142~167 of paragraph 2 on page 4 in the revised manuscript. (PMID:19008921, PMID: 34047789, PMID:8253727, PMID:17244528, PMID: 17612585, PMID:17999726, PMID:11937716, PMID:15634213, PMID:19535733, PMID:17935992, PMID: 16600212, PMID:19571114, PMID:17683935, PMID: 33338748, PMID:32949647, PMID:24290752, PMID:24879158, PMID:16420522, PMID:37821429, PMID:25940348, PMID: 38198575)
Point 3: Abstract, lines 33-34 and 41 in Introduction, Legend to Figure 1, and line 156: ”delivery to autolysosomes for degradation by acidic proteases” Degradation of autophagic cargo requires many types of lysosomal enzymes in addition to proteases (to degrade lipids, carbohydrates, nucleic acids, to remove posttranslational modifications, etc). Please reword all the sentences referring to proteases accordingly.
Response 3: I thank the reviewer for this comment. I have revised “proteases” to “hydrolases” in the revised manuscript. Please see line 33 in the Introduction section on page 1, line 56 in the legend of Figure 1 on page 2, line 101 in the legend of Figure 2 on page 3, and line 169 of paragraph 3 on page 4 in the revised manuscript.
Point 4: In Figure 1, the schematic presentation of microautophagy is problematic. The present figure gives the impression that the large lysosome is internalizing the small endosome as a whole, while the correct presentation would be to indicate that the endosome fuses with the lysosome.
Response 4: I thank the reviewer for the thoughtful comments. The section showing microautophagy in Figure 1 has been revised as the reviewer suggested. Please see Figure 1 on page 2 in the revised manuscript.
Point 5: Line 149-150: “To date, less is known about how autophagosomes fuse with lysosomes to form amphisomes.” Fusion of autophagosome with lysosome makes autolysosome, not amphisome. Should there be ‘how autophagosomes fuse with endosomes’?
Response 5: I appreciate the reviewer for this correction. The “how autophagosomes fuse with lysosomes to form amphisomes” has been corrected to “how autophagosomes fuse with endosomes to form amphisomes”. Please see lines 142~143 of paragraph 2 on page 4 in the revised manuscript.
Point 6: Line 272: ‘Raf family of’ should be ‘Rab family of’
Response 6: I thank the reviewer for this correction. The term “Raf family of” has been corrected to “Rab family of”. Please see line 351 of paragraph 3 on page 9 in the revised manuscript.
Point 7: Line 422: ‘ULK1-mediated YKT6 phosphorylation by the phospho-mimetic mutant (T156E) of YKT6’ should read ‘ULK1-mediated YKT6 phosphorylation and the phospho-mimetic mutant (T156E) of YKT6’
Response 7: I appreciate the reviewer for this comment. The “ULK1-mediated YKT6 phosphorylation by the phospho-mimetic mutant (T156E) of YKT6” has been revised to “ULK1-mediated YKT6 phosphorylation and the phospho-mimetic mutant (T156E) of YKT6”. Please see lines 507~508 of paragraph 3 on page 14 in the revised manuscript.
Point 8: Line 431: SANP29 should be SNAP29
Response 8: I appreciate the reviewer for this correction. The term “SANP29” has been corrected to “SNAP29”. Please see line 533 of paragraph 2 on page 15 and line 830 of paragraph 3 on page 23 in the revised manuscript.
Point 9: Line 580: movment should be movement
Response 9: I thank the reviewer for this correction. The term “movment” has been corrected to “movement”. Please see line 689 of paragraph 1 on page 20 in the revised manuscript.
Point 10: Line 550: Oppostively – I do not know this word. Should it be ‘oppositely’?
Response 10: I appreciate the reviewer for this correction. The term “Oppostively” has been corrected to “oppositely”. Please see line 660 of paragraph 3 on page 19 in the revised manuscript.
